# Towards Defending Multiple Adversarial Perturbations via Gated Batch Normalization

## Abstract

There is now extensive evidence demonstrating that deep neural networks are vulnerable to adversarial examples, motivating the development of defenses against adversarial attacks. However, existing adversarial defenses typically improve model robustness against individual specific perturbation types. Some recent methods improve model robustness against adversarial attacks in multiple $\ell_p$ balls, but their performance against each perturbation type is still far from satisfactory. To better understand this phenomenon, we propose the *multi-domain* hypothesis, stating that different types of adversarial perturbations are drawn from different domains. Guided by the multi-domain hypothesis, we propose *Gated Batch Normalization (GBN)*, a novel building block for deep neural networks that improves robustness against multiple perturbation types. GBN consists of a gated subnetwork and a multi-branch batch normalization (BN) layer, where the gated subnetwork separates different perturbation types, and each BN branch is in charge of a single perturbation type and learns domain-specific statistics for input transformation. Then, features from different branches are aligned as domain-invariant representations for the subsequent layers. We perform extensive evaluations of our approach on MNIST, CIFAR-10, and Tiny-ImageNet, and demonstrate that GBN outperforms previous defense proposals against multiple perturbation types, *i.e.*, $\ell_1$, $\ell_2$, and $\ell_\infty$ perturbations, by large margins of 10-20%.[1]

## 1 Introduction

Deep neural networks (DNNs) have achieved remarkable performance across a wide areas of applications (Krizhevsky et al., 2012; Bahdanau et al., 2014; Hinton et al., 2012), but they are susceptible to *adversarial examples* (Szegedy et al., 2013). These elaborately designed perturbations are imperceptible to humans but can easily lead DNNs to wrong predictions, threatening both digital and physical deep learning applications (Kurakin et al., 2016; Liu et al., 2019a).

To improve model robustness against adversarial perturbations, a number of *adversarial defense* methods have been proposed (Papernot et al., 2015; Engstrom et al., 2018; Goodfellow et al., 2014). Many of these defense methods are based on adversarial training (Goodfellow et al., 2014; Madry et al., 2018), which augment training data with adversarial examples. However, most adversarial defenses are designed to counteract a single type of perturbation (*e.g.*, small $\ell_\infty$-noise) (Madry et al., 2018; Kurakin et al., 2017; Dong et al., 2018). These defenses offer no guarantees for other perturbations (*e.g.*, $\ell_1$, $\ell_2$), and sometimes even increase model vulnerability (Kang et al., 2019; Tramèr & Boneh, 2019). To address this problem, other adversarial training strategies have been proposed with the goal of simultaneously achieving robustness against multiple types of attacks, *i.e.*, $\ell_\infty$, $\ell_1$, and $\ell_2$ attacks (Tramèr & Boneh, 2019; Maini et al., 2020). Although these methods improve overall model robustness against adversarial attacks in multiple $\ell_p$ balls, the performance for each individual perturbation type is still far from satisfactory.

In this work, we propose the multi-domain hypothesis, which states that different types of adversarial perturbation arise in different domains, and thus have separable characteristics. Training on data from multiple domains can be regarded as solving the invariant risk minimization problem (Ahuja et al., 2020), in which an invariant predictor is learnt to achieve the minimum risk for different environments. For a deep learning model, instance-related knowledge can be stored in the weight matrix

---

[1]Our code will be available upon publication.

of each layer, whereas domain-related knowledge can be represented by the batch normalization (BN) layer statistics (Li et al., 2017).

Inspired by the multi-domain hypothesis, we propose to improve model robustness against multiple perturbation types by separating domain-specific information for different perturbation types, and using BN layer statistics to better align data from the mixture distribution and learn domain-invariant representations for multiple adversarial examples types. In particular, we propose a novel building block for DNNs, referred to as *Gated Batch Normalization* (GBN), which consists of a gated sub-network and a multi-branch BN layer. GBN first learns to separate perturbations from different domains on-the-fly and then normalizes them by obtaining domain-specific features. Specifically, each BN branch handles a single perturbation type (*i.e.*, domain). Then, features computed from different branches are aligned as domain-invariant representations that are aggregated as the input to subsequent layers. Extensive experiments on MNIST, CIFAR-10, and Tiny-ImageNet demonstrate that our method outperforms previous defense strategies by large margins, *i.e.*, 10-20%.

## 2 BACKGROUND AND RELATED WORK

In this section, we provide a brief overview of existing work on adversarial attacks and defenses, as well as batch normalization techniques.

### 2.1 ADVERSARIAL ATTACKS AND DEFENSES

Adversarial examples are inputs intentionally designed to mislead DNNs (Szegedy et al., 2013; Goodfellow et al., 2014). Given a DNN $f_\Theta$ and an input image $\mathbf{x} \in \mathbb{X}$ with the ground truth label $\mathbf{y} \in \mathbb{Y}$, an adversarial example $\mathbf{x}_{adv}$ satisfies

$$f_\Theta(\mathbf{x}_{adv}) \neq \mathbf{y} \quad s.t. \quad \|\mathbf{x} - \mathbf{x}_{adv}\| \leq \epsilon,$$

where $\| \cdot \|$ is a distance metric. Commonly, $\| \cdot \|$ is measured by the $\ell_p$-norm ($p \in \{1,2,\infty\}$).

Various defense approaches have been proposed to improve model robustness against adversarial examples (Papernot et al., 2015; Xie et al., 2018; Madry et al., 2018; Liao et al., 2018; Cisse et al., 2017), among which adversarial training has been widely studied and demonstrated to be the most effective (Goodfellow et al., 2014; Madry et al., 2018). Specifically, adversarial training minimizes the worst case loss within some perturbation region for classifiers, by augmenting the training set $\{\mathbf{x}^{(i)}, \mathbf{y}^{(i)}\}_{i=1...n}$ with adversarial examples. However, these defenses only improve model robustness for one type of perturbation (*e.g.*, $\ell_\infty$) and typically offer no robustness guarantees against other attacks (Kang et al., 2019; Tramèr & Boneh, 2019; Schott et al., 2019).

To address this problem, recent works have attempted to improve the robustness against several types of perturbation (Schott et al., 2019; Tramèr & Boneh, 2019; Maini et al., 2020). Schott et al. (2019) proposed Analysis by Synthesis (ABS), which used multiple variational autoencoders to defend $\ell_0$, $\ell_2$, and $\ell_\infty$ adversaries. However, ABS only works on the MNIST dataset. Croce & Hein (2020a) proposed a provable adversarial defense against all $\ell_p$ norms for $p \geq 1$ using a regularization term. However, it is not applicable to the empirical setting, since it only guarantees robustness for very small perturbations (*e.g.*, 0.1 and 2/255 for $\ell_2$ and $\ell_\infty$ on CIFAR-10). Tramèr & Boneh (2019) tried to defend against multiple perturbation types ($\ell_1$, $\ell_2$, and $\ell_\infty$) by combining different types of adversarial examples for adversarial training. Specifically, they introduced two training strategies, "MAX" and "AVG", where for each input image, the model is either trained on its strongest adversarial example or all types of perturbations. More recently, Maini et al. (2020) proposed multi steepest descent (MSD), and showed that a simple modification to standard PGD adversarial training improves robustness to $\ell_1$, $\ell_2$, and $\ell_\infty$ adversaries. In this work, we follow (Tramèr & Boneh, 2019; Maini et al., 2020) to focus on defense against $\ell_1$, $\ell_2$, and $\ell_\infty$ adversarial perturbations, which are the most representative and commonly used perturbations. However, we propose a completely different perspective and solution to the problem.

### 2.2 BATCH NORMALIZATION

BN (Ioffe & Szegedy, 2015) is typically used to stabilize and accelerate DNN training. Let $\mathbf{x} \in \mathbb{R}^d$ denote the input to a neural network layer. During training, BN normalizes each neuron/channel

within $m$ mini-batch data by

$$\hat{\mathbf{x}}_j = BN(\mathbf{x}_j) = \gamma_j \frac{\mathbf{x}_j - \mu_j}{\sqrt{\sigma_j^2 + \xi}} + \beta_j, \ \ j = 1, 2, ..., d, \tag{1}$$

where $\mu_j = \frac{1}{m} \sum_{i=1}^{m} \mathbf{x}_j^{(i)}$ and $\sigma_j^2 = \frac{1}{m} \sum_{i=1}^{m} (\mathbf{x}_j^{(i)} - \mu_j)^2$ are the mini-batch mean and variance for each neuron, respectively, and $\xi$ is a small number to prevent numerical instability. The learnable parameters $\gamma$ and $\beta$ are used to recover the representation capacity. During inference, the population statistics of mean $\hat{\mu}$ and variance $\hat{\sigma}^2$ are used in Eqn. 1, which are usually calculated as the running average over different training iterations $t$ with update factor $\alpha$:

$$\hat{\mu}^t = (1 - \alpha)\hat{\mu}^{t-1} + \alpha\mu^{t-1}, \ \ (\hat{\sigma}^t)^2 = (1 - \alpha)(\hat{\sigma}^{t-1})^2 + \alpha(\sigma^{t-1})^2. \tag{2}$$

A number of normalization techniques have been proposed to improve BN for style-transfer (Huang & Belongie, 2017) and domain adaption (Li et al., 2017; Chang et al., 2019; Deecke et al., 2019). Compared to studies for domain adaptation, our GBN does not require input domain knowledge during inference. In contrast to Mode Normalization (Deecke et al., 2019), which aims to learn a mixture distribution for multi-task learning, GBN separates domain-specific statistics for different input types and then aligns the data to build domain-invariant representations. Recently, Xie & Yuille (2020); Xie et al. (2020) applied separate BNs to clean and adversarial examples for statistics estimation. In contrast to the above two studies that improve image recognition or analyze the properties of BN to adversarial training, we aim to defend against multiple adversarial perturbations. Furthermore, GBN can predict the domain for an input sample and then jointly incorporate the outputs of multiple BN branches to provide the final classification result. Conversely, Xie & Yuille (2020); Xie et al. (2020) manually selected the BN branch for each input image, which requires prior knowledge of whether the input is an adversarial or clean example (the same as domain adaptation).

## 3 GATED BATCH NORMALIZATION

In this section, we present our proposed Gated Batch Normalization (GBN) approach, to improve the model robustness against multiple adversarial perturbations. We first illustrate our multi-domain hypothesis, which states that adversarial examples of different perturbation types are drawn from different domains. Motivated by this hypothesis, we demonstrate the architecture of our GBN block, then describe the training and inference procedures.

### 3.1 MOTIVATION: MULTI-DOMAIN HYPOTHESIS

We assume $N$ *adversarial perturbation types*[2], each characterized by a set $\mathbb{S}^k$ of perturbations for an input $\mathbf{x}$. Let $\mathbb{D}^0$ denote the set of clean examples, and $\mathbb{D}^k (k = 1, ..., N)$ denote the set of adversarial examples generated by the $k$-th adversarial perturbation type $\mathbb{S}^k$. An adversarial example of the $k$-th type $\mathbf{x}_{adv}^k$ is generated by the pixel-wise addition of the perturbation $\delta^k$, *i.e.*, $\mathbf{x}_{adv}^k = \mathbf{x} + \delta^k$. Our *multi-domain hypothesis* states that different types of perturbations $\mathbb{D}^k$ (for all $k \geq 0$, including the clean examples) are drawn from different domains, inspired by the following observations:

(1) Training on a single perturbation type is insufficient for achieving the robustness against other perturbations. Further, training on a mixture of different perturbation types still fail to achieve acceptable performance for each type of perturbation (Tramèr & Boneh, 2019; Maini et al., 2020).

(2) Adversarial examples are separable from clean examples (Metzen et al., 2018). Xie & Yuille (2020) also suggested that clean and adversarial examples are drawn from two different domains.

We first empirically investigate the hypothesis for deep neural networks. Specifically, we train separate BNs for different input domains $\mathbb{D}^k (k \geq 0)$. As illustrated in Figure 1(b), we construct separate mini-batches to estimate BN statistics for different $\mathbb{D}^k$, *i.e.*, $\ell_1$, $\ell_2$, and $\ell_\infty$ adversarial images, as well as for clean images. We defer the implementation details to Appendix A. Figure 1(c) and 1(d) show that different $\mathbb{D}^k$ induce significantly different running statistics, according to the running means and variances of different BN branches.

---

[2]In this work, we consider $N = 3$ adversarial perturbation types: $\ell_1, \ell_2$ and $\ell_\infty$.

We further theoretically demonstrate that for a linear classifier $f(\mathbf{x}) = sign(w^T\mathbf{x} + b)$, different types of adversarial perturbations belong to different domains, with pair-wise Wasserstein distances greater than 0 (*c.f.* Appendix B).

These theoretical and empirical results support our multi-domain hypothesis, and motivate our design of GBN, as discussed below.

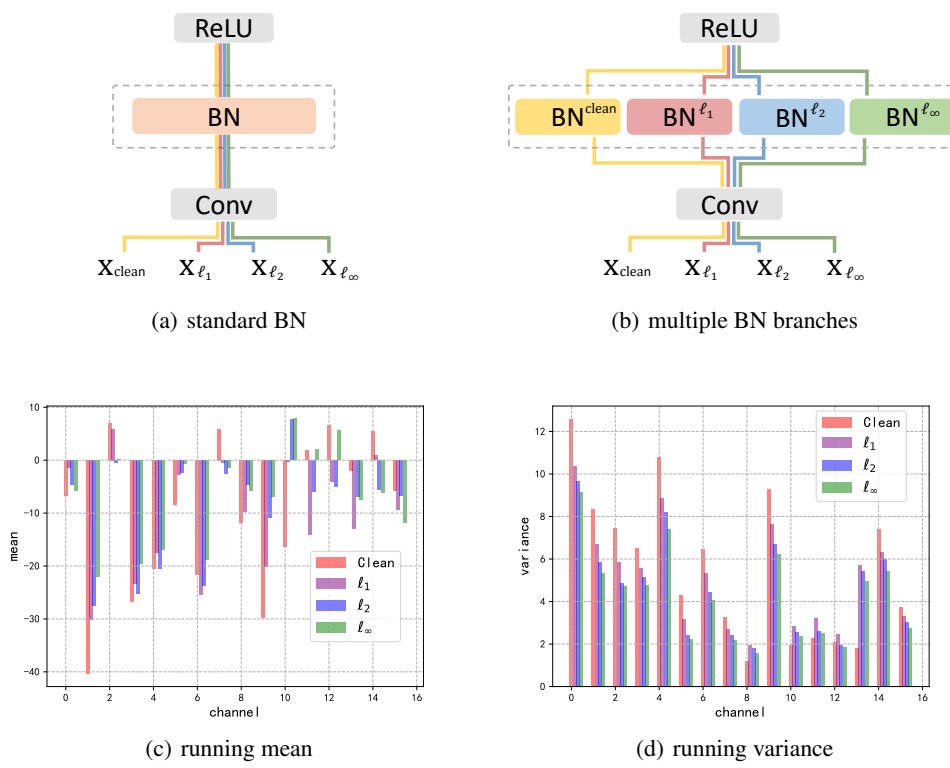

Figure 1: (a) the standard BN structure; (b) the structure with 4 BN branches, which construct different mini-batches for different $\mathbb{D}^k$ to estimate the normalization statistics; (c) and (d): running means and variances of multiple BN branches on 16 randomly sampled channels in a VGG-16's $conv2\_1$ layer, showing that different perturbation types induce different normalization statistics. More details of running statistics of different models can be found in Appendix E.10.

## 3.2 MODEL ARCHITECTURE WITH GATED BATCH NORMALIZATION (GBN)

Training a model that generalizes well on multiple domains can be regarded as solving an invariant risk minimization problem (Ahuja et al., 2020). Define a predictor $f : \mathbb{X} \to \mathbb{Y}$ and the risk achieved by $f$ in domain $\mathbb{D}^k$ as $R_k(f) = \sum_{(\mathbf{x},\mathbf{y})\sim\mathbb{D}^k} \ell(f(\mathbf{x}),\mathbf{y})$, where $\ell(\cdot)$ is the cross entropy loss. We say that a data representation $\pi: \mathbb{X} \to \mathbb{Z}$ elicits an invariant predictor $\omega \circ \pi$ across all domains $k \in \{0, 1, \cdots, N\}$ if there is a classifier $\omega: \mathbb{Z} \to \mathbb{Y}$ that achieves the minimum risk for all domains.

To obtain a domain-invariant representation $\pi$, we use the BN layer statistics to align data from the mixture distribution. Specifically, GBN uses a BN layer with multiple branches $\Psi = \{BN^k(\cdot), k = 0, ..., N\}$ during training, where each branch $BN^k(\cdot)$ is exclusively in charge of domain $\mathbb{D}^k$. The aligned data are then aggregated as the input to subsequent layers to train a classifier $\omega$ which achieves the minimum risk across different domains (*i.e.*, model robust to different perturbations).

One remaining problem is that the model does not know the input domain during inference. To calculate the normalized output in this case, GBN utilizes a gated sub-network $\Phi_\theta(\mathbf{x})$ to predict the domain for each layer input $\mathbf{x}$; we will illustrate how to train $\Phi_\theta(\mathbf{x})$ in Section 3.3. Given the output

of sub-network $\mathbf{g} = \Phi_\theta(\mathbf{x}) \in \mathbb{R}^{N+1}$, we calculate the normalized output in a soft-gated way:

$$\hat{\mathbf{x}} = GBN(\mathbf{x}) = \sum_{k=0}^{N} \mathbf{g}_k \hat{\mathbf{x}}^k, \tag{3}$$

where $\mathbf{g}_k$ represents the confidence of the $k$-th domain for $\mathbf{x}$, and $\hat{\mathbf{x}}^k$ is the normalized output of $BN^k(\cdot)$, which uses the population statistics of the domain $\mathbb{D}^k$ to perform standardization.

We provide an overview of the inference procedure in Figure 2(b), and the details are presented in Algorithm 1. In Appendix E.6, we discussed an alternative approach that takes the top-1 prediction of $\mathbf{g}$ (the hard label), and it achieved comparable performance.

Our GBN aims to disentangle domain-invariant features and domain-specific features: (1) the distributions in different domains are aligned by their normalized outputs (all are standardized distributions), which ensures that the following linear layers learn domain-invariant representations; and (2) the domain-specific features for each domain $\mathbb{D}^k$ are obtained by the population statistics $\{\hat{\mu}^k, \hat{\sigma}^k\}$ of its corresponding BN branch $BN^k(\cdot)$. We show empirically that $\{\hat{\mu}^k, \hat{\sigma}^k\}$ of $BN^k(\cdot)$ learns the domain-specific representation well, as shown in Figure 1.

---

**Algorithm 1** Inference of Gated Batch Normalization (GBN)

**Input:** Layer input $\mathbf{x}$
**Output:** Normalized output by GBN

1: Compute the output of gated sub-network $\mathbf{g} = \Phi_\theta(\mathbf{x})$
2: **for** $k$ in $N+1$ domains **do**
3:     Let $\mathbf{x}$ go through $BN^k(\cdot)$ and obtain the normalized output $\hat{\mathbf{x}}^k$ based on Eqn. 1.
4: **end for**
5: Compute GBN output based on Eqn. 3.

---

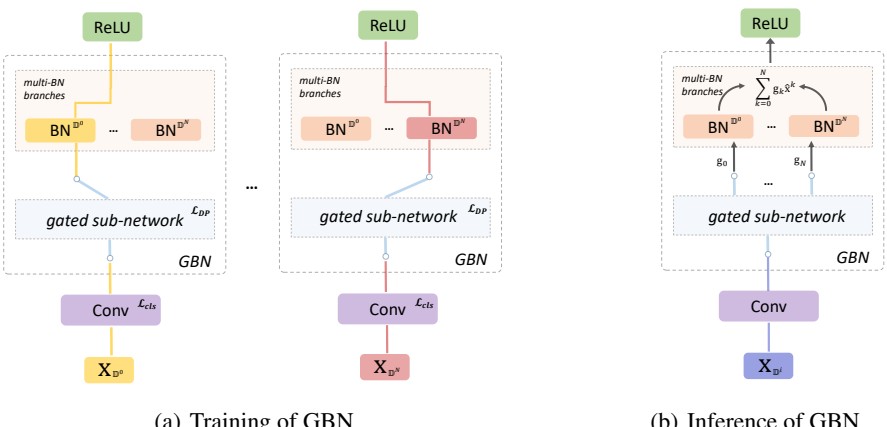

(a) Training of GBN                    (b) Inference of GBN

Figure 2: The training and inference procedures of GBN. During training, samples from different domains (denoted by different colors) are fed into corresponding BN branches to update the running statistics. During inference, given mini-batch data, the gated sub-network first predicts the domain of input $\mathbf{x}$ and then jointly normalizes it using multiple BN branches.

### 3.3 TRAINING

We provide an overview of the training procedure in Figure 2(a), and the details are in Algorithm 2. Specifically, for each mini-batch $\mathcal{B}^0$ consisting of clean samples, we use PGD attack (Madry et al., 2018) to generate batches of adversarial examples $\mathcal{B}^k$ ($k \in \{1, ..., N\}$) for each domain $\mathbb{D}^k$. To capture the domain-specific statistics for different perturbation types, given mini-batch data $\mathcal{B}^k$ from domain $\mathbb{D}^k$, we ensure that $\mathcal{B}^k$ goes through its corresponding BN branch $BN^k(\cdot)$, and we use Eqn. 1

to compute the normalized output $\widehat{\mathcal{B}}^k$. The population statistics $\{\hat{\mu}^k, \hat{\sigma}^k\}$ of $BN^k(\cdot)$ are updated based on Eqn. 2. In other words, we disentangle the mixture distribution for normalization and apply separate BN branches to different perturbation types for statistics estimation.

To train the gated sub-network $\mathbf{g}$, we provide the supervision of the input domain for each training sample. Specifically, we introduce the domain prediction loss $\mathcal{L}_{DP}$:

$$\mathcal{L}_{DP} = \sum_{k=0}^{N} \sum_{(\mathbf{x},k)\in\mathbb{D}^k} \ell(\Phi_\theta(\mathbf{x}), k). \tag{4}$$

Finally, we optimize the parameters $\Theta$ of the entire neural network (*e.g.*, the weight matrices of the convolutional layers, except for the gated sub-network) using the classification loss $\mathcal{L}_{cls}$:

$$\mathcal{L}_{cls} = \sum_{k=0}^{N} \sum_{(\mathbf{x},\mathbf{y})\in\mathbb{D}^k} \ell(f_\Theta(\mathbf{x}; BN^k(\cdot)), \mathbf{y}), \tag{5}$$

**Computational cost.** With respect to the parameters, we have $N+1$ pairs of $\{\hat{\mu}, \hat{\sigma}\}$ to estimate compared to one in standard BN. With respect to the time consumption, in the adversarial learning literature, the most time-consuming procedure in adversarial training is generating adversarial examples. Although we introduce more parameters, our time cost is close to other adversarial defensive baselines (see Appendix E.2 for more details).

---

**Algorithm 2** Training of Gated Batch Normalization (GBN) for Each Iteration.

**Input:** Network $f$ with GBN
**Output:** Robust model parameters $\Theta$ and $\theta$

1: Given mini-batch data $\mathcal{B}^0$, generate corresponding adversarial example batch $\mathcal{B}^k$ ($k \in \{1, ..., N\}$) with specific perturbation type by PGD attacks
2: **for** $k$ in $N+1$ domains **do**
3:    Let $\mathcal{B}^k$ go through $BN^k(\cdot)$ at each layer to obtain normalized outputs $\widehat{\mathcal{B}}^k$ based on Eqn. 1.
4:    Update the population statistics $\{\hat{\mu}^k, \hat{\sigma}^k\}$ of $BN^k(\cdot)$ at each layer based on Eqn. 2.
5: **end for**
6: Update parameters $\theta$ of the gated sub-network at each layer based on Eqn. 4.
7: Update parameters $\Theta$ of the whole network based on Eqn. 5.

---

## 4 EXPERIMENTS

We evaluate the effectiveness of our GBN block to simultaneously achieve robustness against $\ell_1$, $\ell_2$, and $\ell_\infty$ perturbations, which are the most representative and commonly used adversarial perturbation. We conduct experiments on the image classification benchmarks MNIST (LeCun, 1998), CIFAR-10 (Krizhevsky & Hinton, 2009), and Tiny-ImageNet (Wu et al., 2017).

### 4.1 EXPERIMENT SETUP

**Architecture and hyperparameters.** We use LeNet architecture (LeCun et al., 1998) for MNIST; ResNet-20 (He et al., 2016), VGG-16 (Simonyan & Zisserman, 2015), and WRN-28-10 (Zagoruyko & Komodakis, 2016) for CIFAR-10; and ResNet-34 (He et al., 2016) for Tiny-ImageNet. For fair comparisons, we keep the architecture and main hyper-parameters the same for GBN and other baselines. A detailed description of the hyper-parameters can be found in Appendix D.

**Adversarial attacks.** To evaluate the model robustness, we follow existing guidelines (Schott et al., 2019; Tramèr & Boneh, 2019; Maini et al., 2020) and incorporate multiple adversarial attacks for different perturbation types. For MNIST, the magnitude of perturbation for $\ell_1$, $\ell_2$, and $\ell_\infty$ is $\epsilon = 10, 2, 0.3$. For CIFAR-10 and Tiny-ImageNet, the magnitude of perturbation for $\ell_1$, $\ell_2$, and $\ell_\infty$ is $\epsilon = 12, 0.5, 0.03$. For $\ell_1$ attacks, we adopt PGD attack (Madry et al., 2018), and Brendel &

Bethge attack (BBA) (Brendel et al., 2019); for $\ell_2$ attacks, we use PGD attack, C&W attack (Carlini & Wagner, 2017), Gaussian noise attack (Rauber et al., 2017), and boundary attack (BA) (Brendel et al., 2018); For $\ell_\infty$ attacks, we use PGD attack, Fast Gradient Sign Method (FGSM) (Goodfellow et al., 2014), SPSA (Uesato et al., 2018), Nattack (Li et al., 2019), Momentum Iterative Method (MI-FGSM) (Dong et al., 2018), C&W attack, and the recently proposed AutoAttack (Croce & Hein, 2020b) (a stronger ensemble of parameter-free attacks, including APGD, FAB (Croce & Hein, 2020c), and Square Attack (Andriushchenko et al., 2020)). We adopt FoolBox (Rauber et al., 2017) as the implementation for the attacks. Note that, to prove that no obfuscated gradient has been introduced, we adopt both the white-box and the black-box or gradient-free adversarial attacks. A more complete details of all attacks including hyper-parameters can be found in Appendix D.1.

**Adversarial defenses.** We compare with works that achieve robustness against multiple perturbation types. All of them consider the union of $\ell_1, \ell_2, \ell_\infty$ adversaries.[3] We compare with ABS (Schott et al., 2019), MAX, AVG (Tramèr & Boneh, 2019), and MSD (Maini et al., 2020). For completeness, we also compare with TRADES (Zhang et al., 2019) and PGD adversarial training (Madry et al., 2018) with single perturbation type (*i.e.*, $\ell_1, \ell_2$, and $\ell_\infty$, respectively as $P_1, P_2$, and $P_\infty$).

**Normalization techniques.** We further compare our methods to some normalization techniques (*i.e.*, MN (Deecke et al., 2019) and MBN (Xie & Yuille, 2020)). MN extends BN to more than a single mean and variance, detects the modes of the data, and then normalize samples that share common features. For MBN, since manually selecting the BN branch is infeasible in the adversarial setting, we add a 2-way gated sub-network to each MBN block.

## 4.2 Adversarial Robustness for multiple perturbations

In this section, we evaluate model robustness against multiple perturbation types (*i.e.*, $\ell_1, \ell_2$, and $\ell_\infty$) in both the white-box and black-box setting. We use the *worst case* top-1 classification accuracy, where we report the worst result among various attacks for each perturbation type (the higher the better). We also report *all attacks* which represents the worst-case performance computed over all attacks for each example (Maini et al., 2020). Specifically, given an image, if we generate an adversarial attack that leads to wrong predictions using any attacks, we mark it as a failure.

For all attacks except FGSM and $APGD_{CE}$ (the default setting in AutoAttack), we made 5 random restarts for each of the results. As for models using our GBN, we trained each model 3 times, and the results are similar, showing that training with GBN is stable.

The results on MNIST using LeNet, CIFAR-10 using ResNet-20, and Tiny-ImageNet using ResNet-34 are shown in Table 1. We present the detailed breakdown results for individual attacks and more experimental results using VGG-16 and WRN-28-10 on CIFAR-10 in Appendix E.3. We observe that: (1) on multiple perturbation types (*i.e.*, $\ell_1, \ell_2, \ell_\infty$), our GBN consistently outperforms the baselines by large margins of over 10% on all three datasets; (2) due to the trade-off between adversarial robustness and standard accuracy (Tsipras et al., 2019), our clean accuracy is lower than the vanilla model. However, GBN maintains a comparatively high clean accuracy compared to other adversarial defenses; and (3) GBN is easy to train on different datasets, while the other methods (*e.g.*, MSD) are difficult to converge on large datasets such as Tiny-ImageNet.

We also evaluate our model on PGD attacks using different iteration steps (*i.e.*, 50, 100, 200, and 1000) in terms of $\ell_\infty$ norm. As shown in Appendix E.4, our GBN still outperforms other methods and remains a high level of robustness.

**Comparison with other normalization methods.** As shown in Table 1, GBN outperforms MN and MBN by large margins (up to 60%). Here, we provide more comparisons with MN and MBN. Apart from the target task, our GBN differs them on the general ideas: our GBN first aligns the distribution among different perturbations by using one BN branch to handle one perturbation type (learn domain-specific features), and then the aligned distribution contributes to the subsequent modules or layers (*e.g.*, convolutional or fully connected layer) for learning domain-invariant representations. GBN ensure that the normalized output for each perturbation type has a Gaussian distribution, and these Gaussian distributions among different perturbations are well aligned (*e.g.*, one clean example and its corresponding adversarial examples are more likely to be close in the space of normalized output. See Appendix E.9 for the visualization of features after normalization blocks using t-SNE

---

[3]Schott et al. (2019) consider the $\ell_0$ perturbations, which is subsumed within the $\ell_1$ ball of the same radius.

Laurens & Hinton (2008)). By doing this, the other modules are easier to learn the native representation of the data (domain-invariant representations). However, MN and MBN cannot align the distributions. MN aims to preserve the diversity among different distributions, by learning the mixture of Gaussian, not one single Gaussian. As for the modified MBN, the data from different perturbations are mixed in the same BN. The results also confirm our multi-domain hypothesis.

**Different variants of white-box attacks.** Here, we examine the performance of GBN in the more rigorous white-box attack scenario. In this case, we conduct two experiments as follows: (1) attacking the GBN layers, *i.e.*, generate adversarial examples to fool the classification of the gated sub-networks; (2) the adversary manually selects the BN branch to generate adversarial examples. Note that these attacks are specially designed for our defense aiming to provide further analysis and understanding. As shown in Table 15 and 17 in Appendix E.8, GBN remains the same level of robustness against these attacks. We conjecture the reasons are as follows: (1) our GBN does not rely on obfuscated gradients (Athalye et al., 2018) to provide model robustness; (2) different GBN layers use different features to predict the domains, and it is difficult for an adversary to generate perturbations to fool all GBN layers within a model under the magnitude constraint (see Table 16 for more results of prediction accuracy of gated sub-networks at different layers). We defer more discussions to Appendix E.8.

Table 1: Model robustness on different datasets (%). We also provide the Standard Deviation for *All attacks* of each method on each dataset.

(a) LeNet on MNIST

| | Vanilla | TRADES | AVG | MAX | ABS | MSD | MN | MBN | GBN(ours) |
|---|---|---|---|---|---|---|---|---|---|
| $\ell_1$ attacks | 5.2 | 10.1 | 30.9 | 31.8 | / | 41.9 | 24.2 | 60.9 | **73.2** |
| $\ell_2$ attacks | 1.4 | 12.0 | 60.2 | 59.0 | 83.0 | 71.4 | 18.9 | 65.1 | **93.2** |
| $\ell_\infty$ attacks | 0.0 | **77.0** | 45.8 | 58.8 | 17.0 | 38.9 | 0.0 | 19.2 | 69.1 |
| All attacks | 0.0 | 9.8 | 29.7 | 31.2 | 17.0 | 38.9 | 0.0 | 19.2 | **68.9** |
| | ±0.06 | ±0.12 | ±0.22 | ±0.21 | ±0.28 | ±0.24 | ±0.04 | ±0.36 | ±0.30 |
| Clean | **99.3** | 99.1 | 98.7 | 98.2 | 99.0 | 97.3 | 98.1 | 98.7 | 98.7 |

(b) ResNet-20 on CIFAR-10

| | Vanilla | TRADES | $P_1$ | $P_2$ | $P_\infty$ | AVG | MAX | MSD | MN | MBN | GBN(ours) |
|---|---|---|---|---|---|---|---|---|---|---|---|
| $\ell_1$ attacks | 0.0 | 16.1 | 22.3 | 34.2 | 17.9 | 44.9 | 33.3 | 43.7 | 39.8 | 44.9 | **57.7** |
| $\ell_2$ attacks | 0.0 | 59.5 | 1.1 | 51.0 | 50.3 | 59.1 | 56.0 | 58.9 | 30.0 | 20.8 | **68.9** |
| $\ell_\infty$ attacks | 0.0 | 45.0 | 0.0 | 19.3 | 42.1 | 29.2 | 25.1 | 38.0 | 13.2 | 40.1 | **49.9** |
| All attacks | 0.0 | 16.0 | 0.0 | 19.0 | 17.8 | 28.2 | 24.9 | 37.9 | 13.0 | 20.7 | **48.7** |
| | ±0.03 | ±0.32 | ±0.04 | ±0.42 | ±0.38 | ±0.51 | ±0.45 | ±0.58 | ±0.52 | ±0.61 | ±0.66 |
| Clean | **89.7** | 86.9 | 84.3 | 87.9 | 84.2 | 80.6 | 77.0 | 79.1 | 82.3 | 79.4 | 80.7 |

(c) ResNet-34 on Tiny-ImageNet

| | Vanilla | AVG | MAX | MSD | MN | MBN | GBN(ours |
|---|---|---|---|---|---|---|---|
| $\ell_1$ attacks | 5.8 | 27.3 | 17.0 | 7.6 | 8.6 | 36.6 | **44.2** |
| $\ell_2$ attacks | 10.3 | 29.1 | 24.2 | 11.0 | 17.9 | 31.1 | **42.1** |
| $\ell_\infty$ attacks | 0.0 | 4.5 | 2.7 | 5.1 | 6.8 | 19.4 | **38.7** |
| All attacks | 0.0 | 4.6 | 2.2 | 5.0 | 6.7 | 20.0 | **38.9** |
| | ±0.06 | ±0.65 | ±0.58 | ±0.69 | ±0.63 | ±0.52 | ±0.62 |
| Clean | **54.2** | 41.6 | 36.3 | 28.8 | 46.7 | 45.7 | 43.4 |

## 4.3 Ablation Study

**Attacks outside the perturbation model.** To examine whether GBN could generalize to perturbation types that the model was not trained on, we trained a variant of GBN models that only includes 3 BN branches. Specifically, these GBN models are trained on two $\ell_p$ perturbation types only, but are evaluated on all the three $\ell_p$ perturbations, including the held-out perturbation that is not trained. Unsurprisingly, compared to the full GBN model with 4 BN branches, the robustness of the held-out perturbation type decreases to some degree. However, the robustness is still significantly better than the vanilla model, and sometimes even outperforms other baseline defenses that are trained on all perturbation types (*c.f.* Appendix E.7).

**Adding GBN into different layers.** Our GBN block can be inserted at any layer in a deep network. In this part, we study GBN by single-layer and layer-group studies. We first add GBN to different single layers: as shown in Figure 3(a), the standard performance and adversarial robustness decrease as we add GBN into the deeper layers; we then add GBN to layer groups (*i.e.*, top-$m$ layers): according to the results in Figure 3(b), the adversarial robustness improves (but the clean accuracy remains comparatively stable) as more layers are involved. In summary, adding GBN into shallow layers achieves better performance with respect to both clean and adversarial accuracy. The reasons for this might be two-fold: (1) as shown in (Liu et al., 2019b), shallow layers are more critical to model robustness than deeper layers; (2) as the layer depth increases, the features from different domains are highly entangled, making it harder for models to separate them (*c.f.* Appendix E.10.).

**Predictions of the gated sub-network** Moreover, we provide the prediction results of the gated sub-network in GBN, *i.e.*, the classification accuracy for the domain of different input samples. As shown in Table 11, for MNIST, the gated sub-network of LeNet at different layers achieves high prediction accuracy for the input domains (*i.e.*, clean, $\ell_1$, $\ell_2$, and $\ell_\infty$ adversarial examples). However, on CIFAR-10, the prediction accuracy of the gated sub-network drops with the increasing of the layer depth. We can conclude that shallow layers are more critical to model robustness as the prediction accuracy at the shallow layers is higher than that of the deeper layers. The reason might be that shallow layers contain limited information about the image statistics and can be learned via a few of samples (Asano et al., 2020), in which the features are more easy to separate. However, the reason for the indistinguishability of features in the first layer is still unexplored, which we leave for future works. To further improve model robustness against multiple perturbations, we suggest to improve the prediction accuracy of the gated sub-network.

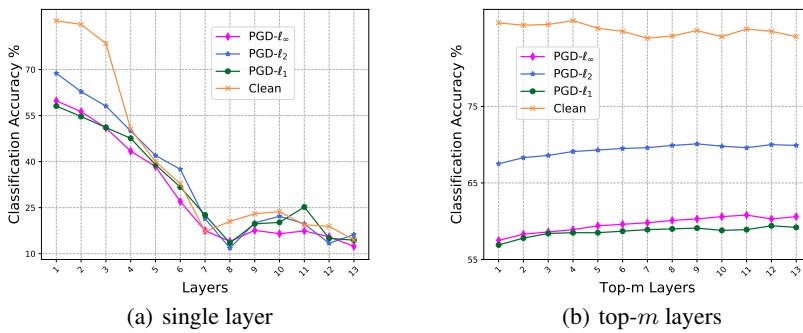

| (a) single layer | (b) top-$m$ layers |

Figure 3: (a) demonstrates the results of adding GBN to different single layers. (b) shows the results of adding GBN to top-$m$ layers. All the experiments are conducted using a VGG-16 on CIFAR-10.

## 5    CONCLUSIONS

Most adversarial defenses are typically tailored to a single perturbation type (*e.g.*, small $\ell_\infty$-noise), but offer no guarantees against other attacks. To better understand this phenomenon, we explored the *multi-domain* hypothesis that different types of adversarial perturbations are drawn from different domains. Guided by this hypothesis, we propose a novel building block for DNNs, *Gated Batch Normalization* (GBN), which consists of a gated network and a multi-branches BN layer. The gated sub-network separates different perturbation types, and each BN branch is in charge of a single perturbation type and learns the domain-specific statistics for input transformation. Then, features from different branches are aligned as domain-invariant representations for the subsequent layers. Extensive experiments on MNIST, CIFAR-10, and Tiny-ImageNet demonstrate that our method outperforms previous proposals against multiple perturbation types by large margins of 10-20%.

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

## A    EXPERIMENTAL SETTING FOR SEPARATING BNS

We provide the experimental setting for separating BNs study in Section 3.1. Specifically, we train a DNN on CIFAR-10 with each BN layer composed of 4 branches (*i.e.*, we replace each BN with 4 BN branches). During training, we construct different mini-batch for clean, $\ell_1$, $\ell_2$, and $\ell_\infty$ adversarial images to estimate the normalization statistics of each branch of BN; during inference, given a mini-batch of data from a specific domain, we manually activate the corresponding BN branch and deactivate the others. We update the running statistics of each BN branch separately but optimize the model parameters using the sum of the four losses.

## B    PROOF IN SECTION 3.1

*Proof.* We first calculate the optimal perturbation $\delta^p$ in the $l_p$ ball of radius $\epsilon$. For simplicity, we assume that the classification loss is given by $l(f(\mathbf{x}), \mathbf{y}) = h(-\mathbf{y}f(\mathbf{x}))$ where $h$ is a non-decreasing function. Note that this is a broad family of losses including the hinge loss and logistic loss. The objective of the adversary is to find a perturbation $\delta^p$ which maximizes the loss. For the linear classifier, $\delta^p$ can be explicitly calculated as follows

$$
\begin{aligned}
\delta^p &= \arg\max_{||\delta||_p \le \epsilon} l(f(\mathbf{x}+\delta), \mathbf{y}) = \arg\max_{||\delta||_p \le \epsilon} h(-\mathbf{y}f(\mathbf{x}+\delta)) \\
&= \arg\max_{||\delta||_p \le \epsilon} -\mathbf{y}w^T\delta \\
&= \epsilon \cdot \arg\max_{||\delta||_p \le 1} -\mathbf{y}w^T\delta \\
&= \epsilon\partial||-\mathbf{y}w||_q,
\end{aligned}
$$

where $1/p + 1/q = 1$ and the last equality uses the property that $\partial||x||_q = \arg\max_{||s||_p \le 1} s^T\mathbf{x}$. Notice that the optimal perturbation $\delta^p$ is independent of the inputs corresponding to a particular label. Therefore, the new distribution $\mathbb{D}_{\mathbf{y}}^p$ of the adversarial examples generated from the perturbation type $S^p$ can be written as $\mathbb{D}_{\mathbf{y}}^0 + \epsilon\partial||-\mathbf{y}w||_q$, where $\mathbb{D}^0$ represents the clean example domain.

For $p = 1$ or $p = \infty$, it is easy to verify that $\partial||-\mathbf{y}w||_q = (0, 0, \cdots, sign(-\mathbf{y}w_i), \cdots, 0)$ where $i = \arg\max_j |w_j|$ or $\partial||-\mathbf{y}w||_q = (sign(-\mathbf{y}w_1), \cdots, sign(-\mathbf{y}w_j), \cdots, sign(-\mathbf{y}w_d))$ respectively.

For $p = 2$, we can simply obtain that $\partial||-\mathbf{y}w||_2 = (-\mathbf{y}w_i/||w||_2, \cdots, -\mathbf{y}w_d/||w||_2)$.

Thus, we have the Wasserstain distance between $\mathbb{D}_{\mathbf{y}}^{\ell_1}$ and $\mathbb{D}_{\mathbf{y}}^{\ell_\infty}$ as

$$
W_2(\mathbb{D}_{\mathbf{y}}^{\ell_1}, \mathbb{D}_{\mathbf{y}}^{\ell_\infty}) = W_2(\mathbb{D}_{\mathbf{y}}^0 + \epsilon\partial||-\mathbf{y}w||_\infty, \mathbb{D}_y^0 + \epsilon\partial||-\mathbf{y}w||_1) = \epsilon\sqrt{d-1},
$$

where the last equality follows from the translation invariance of Wasserstein distance. Similarly, the Wasserstein distance between $\mathbb{D}_{\mathbf{y}}^{\ell_2}$ and $\mathbb{D}_{\mathbf{y}}^{\ell_1}$ is

$$
W_2(\mathbb{D}_{\mathbf{y}}^{\ell_1}, \mathbb{D}_{\mathbf{y}}^{\ell_2}) = \epsilon\sqrt{2 - 2|w_i|/||w||_2}.
$$

Finally, the Wasserstein distance between $\mathbb{D}_{\mathbf{y}}^{\ell_2}$ and $\mathbb{D}_{\mathbf{y}}^{\ell_\infty}$ can be calculated as

$$
W_2(\mathbb{D}_{\mathbf{y}}^{\ell_2}, \mathbb{D}_{\mathbf{y}}^{\ell_\infty}) = \epsilon\sqrt{d + 1 - 2||w||_1/||w||_2}.
$$

$\square$

## C    GBN IMPLEMENATION DETAILS

**Network architecture.** We first illustrate the architecture of the gated sub-network $\mathbf{g}$ in GBN. As shown in Figure 4, we devise two types of gated sub-network, namely *Conv gate* and *FC gate*.

The notation Conv2d($d, s, k$) refers to a convolutional layer with $k$ filters whose size are $d \times d$ convolved with stride $s$. The notation ReLU($\cdot$) denotes the rectified linear unit used as the activation function in the network. FC($m$) denotes a fully connected layer with output size $m$.

*Conv gate* is a convolutional neural network consisting of two convolutional layers, two ReLU layers and one fully-connected layer. $c$ denotes the input channel of this gate. In the default setting, we use stride 1 for the first convolutional layer and 2 for the second. For the filter sizes in both layers,

padding is set to 3 and 1, respectively. We set the fully-connected layer output size $m$=4 to match the four data distributions used in our experiment.

*FC gate* is a simple fully-connected neural network that includes two fully-connected layers and one ReLU layer. We use $m$=512 and $m$=4 for the two fully-connected layers.

```
Conv2d(3, 1, c)
    ReLU()                      FC(512)
Conv2d(3, 2, 2 * c)             ReLU()
    ReLU()                       FC(4)
     FC(4)                     Softmax()
   Softmax()
```

(a) Conv gate          (b) FC gate

Figure 4: The architecture of the gated sub-network used in this paper.

To train models containing GBN, we add the GBN block into all layers within a model. Specifically, we set the **g** in the first GBN as the *Conv gate* and use the *FC gates* for other GBN blocks to further capture domain-specific information and improve model robustness. We use the default values for $\xi$ and $\alpha$ in PyTorch.

**Discussion.** To empirically prove the effectiveness of the above strategy, we conduct additional experiments using the *Conv gate* and *FC gate* for all GBN blocks. In other words, we train a VGG-16 model with all GBN blocks using the *Conv gate* denoted "$\text{Conv}_{all}$", and train another model with all GBN blocks using the *FC gate* denoted "$\text{FC}_{all}$". As shown in Table 2, our strategy (denoted "Conv+FC") achieves the greatest robustness. Thus, we use *Conv gate* for all the GBN blocks in the single layer study, and use *Conv gate* for the first GBN and *FC gates* for the other GBN blocks in the layer group study (Section 4.3).

We conjecture that there are two reasons for this: (1) the running statistics between different domains in the first layer are almost indistinguishable (*c.f.* Appendix E.10). Thus, solely using the FC layer (*FC gate*) fails to extract sufficient features from the first layer to perform correct classification; (2) using conv layers for all GBN blocks may suffer from over-fitting problem, since only adding GBN into the first layer achieves considerable robustness (as shown in Figure 3(a)). We will further address it in the future studies.

Table 2: Model robustness of VGG-16 on CIFAR-10 (the higher the better).

|  | $\text{Conv}_{all}$ | $\text{FC}_{all}$ | Conv+FC |
| --- | --- | --- | --- |
| PGD-$\ell_1$ | 30.9% | 35.5% | 58.5% |
| PGD-$\ell_2$ | 29.9% | 34.6% | 69.5% |
| PGD-$\ell_\infty$ | 20.6% | 32.6% | 60.1% |
| Clean accuracy | 41.6% | 39.6% | 80.7% |

## D  EXPERIMENTAL SETTINGS

**Dataset details.** MNIST is a dataset containing 10 classes of handwritten digits of size $28 \times 28$ with 60,000 training examples and 10,000 test instances. CIFAR-10 consists of 60,000 natural scene color images in 10 classes of size $32 \times 32$. Tiny-Imagenet has 200 classes of size $64 \times 64$, and each class contains 500 training images, 50 validation images, and 50 test images.

**Implementation details.** Here, we provide the details of the experimental settings for GBN and other baselines on different datasets. All the models for each method on MNIST, CIFAR-10, and Tiny-ImageNet are trained for 40, 40, and 20 epochs, respectively. We set the mini-batch size=64, use the SGD optimizer with weight decay 0.0005 for Tiny-ImageNet and use no weight decay for MNIST and CIFAR-10. We set the learning rate as 0.1 for MNIST and 0.01 for CIFAR-10 and

Tiny-ImageNet. For these baselines, we use the published implementations for ABS[4], MAX/AVG[5], MSD[6], and TRADES[7].

## D.1 ADVERSARIAL ATTACKS

In this paper, we adopt both the white-box and black-box adversarial attacks. In the white-box setting, adversaries have the complete knowledge of the target model and can fully access the model; while in the black-box setting, adversaries have limited knowledge of the target classifier (*e.g.*, its architecture) but can not access the model weights.

For $\ell_1$ attacks, we use PGD attack (Madry et al., 2018), and Brendel & Bethge attack (BBA) (Brendel et al., 2019).

For $\ell_2$ attacks, we use PGD attack (Madry et al., 2018), C&W attack (Carlini & Wagner, 2017), Gaussian noise attack (Rauber et al., 2017), and boundary attack (BA) (Brendel et al., 2018).

For $\ell_\infty$ attacks, though $\ell_\infty$ PGD adversary is quite effective, for completeness, we additionally use attacks including, Fast Gradient Sign Method (FGSM) (Goodfellow et al., 2014), SPSA (Uesato et al., 2018), Nattack (Li et al., 2019), Momentum Iterative Method (MI-FGSM) (Dong et al., 2018), and AutoAttack (Croce & Hein, 2020b) (a stronger ensemble of parameter-free attacks, including APGD, FAB (Croce & Hein, 2020c), and Square Attack (Andriushchenko et al., 2020)) .

Among the attacks mentioned above, BA, SPSA, Nattack, and Square Attack are black-box attacks. We then provide the hyper-parameters of each attack.

**PGD-$\ell_1$.** On MNIST, we set the perturbation magnitude $\epsilon$=10, iteration number $k$=50, and step size $\alpha$=$\epsilon$/10. On CIFAR-10 and Tiny-ImageNet, we set the perturbation magnitude $\epsilon$=12, iteration number $k$=50, and step size $\alpha$=0.05.

**PGD-$\ell_2$.** On MNIST, we set the perturbation magnitude $\epsilon$=2.0, iteration number $k$=100, and step size $\alpha$=0.1. On CIFAR-10 and Tiny-ImageNet, we set the perturbation magnitude $\epsilon$=0.5, iteration number $k$=50, and step size $\alpha$=$\epsilon$/10.

**PGD-$\ell_\infty$.** On MNIST, we set the perturbation magnitude $\epsilon$=0.3, iteration number $k$=50, and step size $\alpha$=0.01. On CIFAR-10 and Tiny-ImageNet, we set the perturbation magnitude $\epsilon$=0.03, iteration number $k$=40, and step size $\alpha$=$\epsilon$/10.

**PGD-1000-$\ell_\infty$.** On CIFAR-10, we set the perturbation magnitude $\epsilon$=0.03, iteration number $k$=1000, and step size $\alpha$=$\epsilon$/10.

**BBA**. On MNIST, CIFAR-10 and Tiny-ImageNet, we set the perturbation magnitude $\epsilon$=10, 12, 12 in terms of $\ell_1$ norm, respectively. For all datasets, we set the number of optimization steps as 1000, learning rate as 0.001, momentum as 0.8, and the binary search steps as 10.

**C&W-$\ell_2$.** On MNIST, CIFAR-10 and Tiny-ImageNet, we set the perturbation magnitude $\epsilon$=2.0, 0.5, 0.5 in terms of $\ell_2$ norm, respectively. For all datasets, we set the number of optimization steps as 10000, each step size as 0.01, and the confidence required for an example to be marked as adversarial as 0.

**C&W-$\ell_\infty$.** On CIFAR-10, we set the perturbation magnitude $\epsilon$=0.03 in terms of $\ell_\infty$ norm. We set the number of optimization steps as 10000, each step size as 0.01, and the confidence required for an example to be marked as adversarial as 0.

**Gaussian noise**. On MNIST, CIFAR-10 and Tiny-ImageNet, we set the perturbation magnitude $\epsilon$=2.0, 0.5, 0.5 in terms of $\ell_2$ norm, respectively.

**BA**. On MNIST, CIFAR-10 and Tiny-ImageNet, we set the perturbation magnitude $\epsilon$=2.0, 0.5, 0.5 in terms of $\ell_2$ norm, respectively. For all datasets, we set the maximum number of steps as 25000, initial step size for the orthogonal step as 0.01, and initial step size for the step towards the target as 0.01.

---

[4] https://github.com/bethgelab/AnalysisBySynthesis
[5] https://github.com/ftramer/MultiRobustness
[6] https://github.com/locuslab/robust_union
[7] https://github.com/yaodongyu/TRADES

**FGSM**. On MNIST, we set the perturbation magnitude $\epsilon$=0.3 in terms of $\ell_\infty$ norm. On CIFAR-10 and Tiny-ImageNet, we set the perturbation magnitude $\epsilon$=0.03 in terms of $\ell_\infty$ norm.

**MI-FGSM**. On MNIST, we set the perturbation magnitude $\epsilon$=0.3, the decay factor $\mu$=1 in terms of $\ell_\infty$ norm, and set the step number $k$=10. On CIFAR-10 and Tiny-ImageNet, we set the perturbation magnitude $\epsilon$=0.03, the decay factor $\mu$=1 in terms of $\ell_\infty$ norm, and set the step number $k$=10. The step size $\alpha=\epsilon/k$.

**SPSA**. We set the maximum iteration as 100, the batch size as 8192, and the learning rate as 0.01. The magnitude of perturbation is 0.3 in terms of $\ell_\infty$ norm on MNIST, and 0.03 in terms of $\ell_\infty$ norm on CIFAR-10 and Tiny-ImageNet.

**NATTACK**. We set $T$=600 as the maximum number of optimization iterations, $b$=300 for the sample size, variance of the isotropic Gaussian $\sigma^2$=0.01, and learning rate as 0.008. The magnitude of perturbation is 0.3 in terms of $\ell_\infty$ norm on MNIST, and 0.03 in terms of $\ell_\infty$ norm on CIFAR-10 and Tiny-ImageNet.

**AutoAttack**. AutoAttack selects the following variants of adversarial attacks: $\text{APGD}_{CE}$ without random restarts, $\text{APGD}_{DLR}$, the targeted version of FAB as $\text{FAB}^T$, and Square Attack with one run of 5000 queries. We use 100 iterations for each run of the white-box attacks. For APGD, we set the momentum coefficient $\alpha$=0.75, $\rho$=0.75, initial step size $\eta^{(0)}$=2$\epsilon$, where $\epsilon$ is the perturbation magnitude in terms of the $\ell_\infty$ norm. For FAB, we keep the standard hyper-parameters based on the implementation of AdverTorch. For Square Attack, we set the initial value for the size of the squares p = 0.8.

## D.2 Adversarial Defenses

**ABS.** ABS uses multiple variational autoencoders to construct a complex generative architecture to defend against adversarial examples in the MNIST dataset. As for the limited code released by the authors, we directly use the results reported in Schott et al. (2019). They set the perturbation magnitude $\epsilon$=12, 1.5, 0.3 for $\ell_0$, $\ell_2$, and $\ell_\infty$ attack, respectively. ABS used an $\ell_0$ perturbation model of a higher radius and evaluated against $\ell_0$ attacks. So the reported number is a near estimate of the $\ell_1$ adversarial accuracy.

**AVG.** For each batch of clean data (size=64), we generate corresponding adversarial examples ($\ell_1$, $\ell_2$, and $\ell_\infty$) using PGD attack. We train the model using a combination of clean, $\ell_1$, $\ell_2$, and $\ell_\infty$ adversarial examples simultaneously. The hyper-parameters of PGD adversaries can be found in Appendix D.1 (PGD-$\ell_1$, PGD-$\ell_2$, and PGD-$\ell_\infty$).

**MAX.** For each batch of clean data (size=64), we generate the strongest adversarial examples (one of the $\ell_1$, $\ell_2$, and $\ell_\infty$ attack) using PGD attack. We train the model using a combination of clean examples and the strongest attack. The hyper-parameters of PGD adversaries can be found in Appendix D.1 (PGD-$\ell_1$, PGD-$\ell_2$, and PGD-$\ell_\infty$).

**MSD.** MSD creates a single adversarial perturbation by simultaneously maximizing the worst-case loss over all perturbation models at each projected steepest descent step. For MNIST, we set the perturbation magnitude $\epsilon$=10, 2.0, 0.3 for $\ell_1$, $\ell_2$, and $\ell_\infty$ attack, respectively, and iteration number $k$=100. For CIFAR-10 and Tiny-ImageNet, we set the perturbation magnitude $\epsilon$=12, 0.5, 0.03 for $\ell_1$, $\ell_2$, and $\ell_\infty$ attack respectively, and iteration number $k$=50.

**TRADES.** TRADES is an adversarial defense method trading adversarial robustness off against accuracy which won 1st place in the NeurIPS 2018 Adversarial Vision Challenge. We set $1/\lambda$=3.0 and set other hyper-parameters as the default values following the original paper.

$P_1$. We adversarially train the model with 50% clean examples and 50% adversarial examples generated by $\ell_1$ PGD attack for each mini-batch data. For PGD attack, on CIFAR-10, we set the perturbation magnitude $\epsilon$=12, iteration number $k$=50, and step size $\alpha$=0.05.

$P_2$. We adversarially train the model with 50% clean examples and 50% adversarial examples generated by $\ell_2$ PGD attack for each mini-batch data. For PGD attack, on CIFAR-10, we set the perturbation magnitude $\epsilon$=0.5, iteration number $k$=50, and step size $\alpha=\epsilon/10$.

$P_\infty$. We adversarially train the model with 50% clean examples and 50% adversarial examples generated by $\ell_\infty$ PGD attack for each mini-batch data. For PGD attack, on CIFAR-10, we set the perturbation magnitude $\epsilon$=0.03, iteration number $k$=40, and step size $\alpha$=$\epsilon$/10.

### D.3 NORMALIZATION TECHNIQUES

**MN**. We set the number of modes in MN to 2, which achieves the best performance according to the original paper (Deecke et al., 2019). During training, we feed the model with a mixture of clean, $\ell_1$, $\ell_2$, and $\ell_\infty$ adversarial examples using the same setting as AVG.

**MBN**. The original MBN (Xie et al., 2020; Xie & Yuille, 2020) manually selects the BN branches for clean and adversarial examples during inference, which is infeasible in adversarial defense setting (the model is unaware of the type of inputs). Thus, we add the 2-way gated sub-network in MBN to predict the input domain label; we then keep the following 2 BN branches the same. During training, we compel the clean examples to go through the first BN branch and the adversarial examples (*i.e.*, $\ell_1$, $\ell_2$, and $\ell_\infty$) to the second BN branch. The adversarial examples are generated via PGD using the same setting as AVG.

## E MORE EXPERIMENTAL RESULTS

In this section, we provide further experimental results.

### E.1 MULTIPLE PERTURBATIONS FROM THE FOURIER PERSPECTIVE

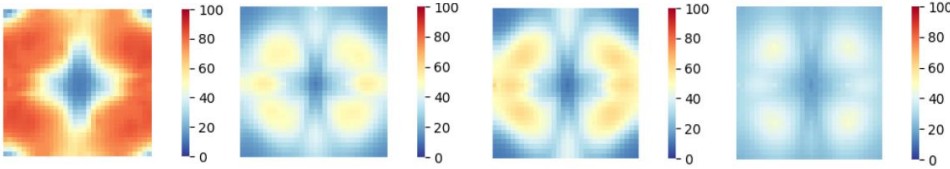

Figure 5: Model sensitivity to additive noise aligned with different Fourier basis vectors on CIFAR-10. From left to right: vanilla, $\ell_1$-trained, $\ell_2$-trained, and $\ell_\infty$-trained models. The numbers indicate the model error rates.

We study the differences between different perturbation types from the Fourier perspective (Yin et al., 2019). We evaluate four different ResNet-20 models: a vanilla model and 3 adversarially-trained models using only $\ell_1$, $\ell_2$, and $\ell_\infty$ adversarial examples, respectively. We visualize the Fourier heatmap following (Yin et al., 2019) and shift the low frequency components to the center of the spectrum. Error rates are averaged over 1000 randomly sampled images from the test set on CIFAR-10. The redder the zone, the higher the error rate of the model to perturbations of specific frequency. As shown in Figure 5, models trained using different $\ell_p$ perturbations demonstrate different model weaknesses in the Fourier heatmap (*i.e.*, different hot zones). For example, $\ell_2$-PAT is more susceptible to high-frequency perturbations while less susceptible to some low-frequency perturbations (the smaller blue zone in the center), while $\ell_\infty$-PAT is comparatively more robust to middle-frequency perturbations (light yellow zones). Therefore, different perturbation types have different frequency properties, and models trained on different perturbations are sensitive to noise from different frequencies. This observation further demonstrates that different perturbation types may arise from different domains (our multi-domain hypothesis).

### E.2 COMPUTATIONAL COST

Here, we provide the time consumption of AVG, MAX, MSD, MN, MBN, and our GBN. All the experiments are conducted on a NVIDIA Tesla V100 GPU cluster. We compute the overall training time using ResNet-20 on CIFAR-10 for 40 epochs. We compute the inference time using ResNet-20

on CIFAR-10 for 10000 images. As seen in Table 3, our method achieves comparable results to other baselines, demonstrating its applicability in practice.

Table 3: Runtime analysis of different methods on CIFAR-10 (the lower the better).

|  | AVG | MAX | MSD | MN | MBN | **GBN(ours)** |
|---|---|---|---|---|---|---|
| Training speed (seconds/ epoch) | 1837.9 | 1113.2 | 689.8 | 19329.1 | 2737.9 | 2802.6 |
| Training duration (hours) | 20.4 | 12.4 | 7.7 | 214.8 | 30.4 | 31.1 |
| Inference speed (seconds/ 10000 images) | 0.60 | 0.58 | 0.61 | 4.8 | 1.9 | 2.0 |

### E.3 ADVERSARIAL ROBUSTNESS AGAINST MULTIPLE PERTURBATIONS

In this part, we provide the breakdown for each individual attack on MNIST, CIFAR-10, and Tiny-ImageNet in Tabel 4, 5, and 6. We also report the results for individual attacks of AutoAttack in 9. Further, we provide the breakdown for each individual attack on CIFAR-10 using VGG-16 and WideResNet-28-10 in Table 7 and Table 8.

According to the results, our GBN outperforms other methods for almost all attacks by large margins. However, it is reasonable to notice that our GBN shows slightly weaker or comparable performance on some individual attacks compared to defenses trained for the specific perturbation types. For example, $P_1$ outperforms GBN for PGD-$\ell_1$ on CIFAR-10 and TRADES shows better performance for some $\ell_\infty$ attacks on MNIST.

Table 4: Model robustness of LeNet on MNIST over each individual attack (the higher the better).

|  |  | Vanilla | TRADES | AVG | MAX | ABS | MSD | MN | MBN | **GBN** |
|---|---|---|---|---|---|---|---|---|---|---|
| $\ell_1$ attacks | PGD-$\ell_1$ | 74.4% | 80.8% | 77.9% | 77.3% | / | 77.0% | 80.4% | 79.6% | **86.8%** |
|  | BBA | 6.5% | 10.2% | 31.2% | 33.1% | / | 42.6% | 25.2% | 64.5% | **79.6%** |
| $\ell_2$ attacks | PGD-$\ell_2$ | 24.7% | 90.1% | 73.5% | 74.6% | / | 69.5% | 77.2% | 78.5% | **97.8%** |
|  | C&W-$\ell_2$ | 2.0% | 42.5% | 68.1% | 67.6% | / | 72.4% | 20.0% | 67.3% | **98.1%** |
|  | Gaussian Noise | 99.2% | 98.5% | 98.7% | 98.2% | 98.0% | 97.3% | 97.6% | 98.6% | **99.2%** |
|  | BA | 10.5% | 12.1% | 60.9% | 59.2% | 83.0% | 78.7% | 24.2% | 90.2% | **97.7%** |
| $\ell_\infty$ attacks | PGD-$\ell_\infty$ | 64.2% | 96.1% | 72.6% | 74.7% | / | 51.6% | 78.9% | 80.3% | **96.3%** |
|  | FGSM | 48.7% | 96.9% | 87.5% | 86.3% | 34.0% | 69.7% | **97.5%** | 75.4% | 89.5% |
|  | MI-FGSM | 36.2% | **96.8%** | 82.1% | 81.5% | 17.0% | 60.9% | 90.7% | 40.7% | 85.2% |
|  | SPSA | 37.6% | 93.3% | 64.2% | 60.5% | / | 71.4% | 77.8% | 77.8% | **98.2%** |
|  | NATTACK | 40.4% | 91.7% | 79.5% | 78.9% | / | 85.2% | 90.2% | 89.3% | **97.7%** |
|  | AutoAttack | 0.0% | **77.1%** | 46.1% | 60.1% | / | 39.0% | 0.0% | 20.0% | 71.7% |
| All attacks | - | 0.0% | 9.8% | 29.7% | 31.2% | 17.0% | 38.9% | 0.0% | 19.2% | **68.9%** |
| Clean accuracy | - | **99.3%** | 99.1% | 98.7% | 98.2% | 99.0% | 97.3% | 98.1% | 98.7% | 98.7% |

In summary, our proposed GBN trains robust models in terms of multiple perturbation types (*i.e.*, $\ell_1, \ell_2, \ell_\infty$) and outperforms other methods by large margins.

### E.4 ATTACKS WITH DIFFERENT ITERATION STEPS

Here, we further provide the experimental results of PGD adversarial attacks using different iteration step numbers (*i.e.*, 50, 100, 200, 1000) in terms of $\ell_\infty$ norm on CIFAR-10. For PGD attacks, we set the perturbation magnitude $\epsilon$=0.03, step size $\alpha$=$\epsilon$/10, and different iteration steps $k$. As shown in Table 10, our GBN outperforms other methods on PGD attacks using different iteration steps.

Table 5: Model robustness of ResNet-20 on CIFAR-10 over each individual attack (the higher the better).

| | | Vanilla | TRADES | $P_1$ | $P_2$ | $P_\infty$ | AVG | MAX | MSD | MN | MBN | **GBN** |
|---|---|---|---|---|---|---|---|---|---|---|---|---|
| $\ell_1$ attacks | PGD-$\ell_1$ | 31.7% | 25.7% | **70.3%** | 39.8% | 18.7% | 47.4% | 38.1% | 49.9% | 44.1% | 47.7% | 58.5% |
| | BBA | 0.0% | 16.2% | 22.4% | 34.2% | 18.0% | 45.0% | 33.4% | 43.8% | 40.0% | 45.8% | **69.1%** |
| $\ell_2$ attacks | PGD-$\ell_2$ | 32.1% | 59.8% | 19.8% | 62.8% | 60.4% | 59.2% | 58.0% | 62.8% | 49.4% | 60.5% | **69.5%** |
| | C&W-$\ell_2$ | 0.0% | 59.5% | 4.8% | 51.2% | 50.4% | 59.7% | 56.0% | 62.5% | 30.1% | 20.8% | **74.9%** |
| | Gaussian Noise | 81.2% | 79.8% | 68.9% | 84.8% | 81.6% | 76.4% | 62.9% | 69.1% | 60.2% | 49.5% | 79.1% |
| | BA | 0.3% | 60.2% | 1.2% | 62.3% | 58.4% | 62.0% | 57.8% | 59.9% | 39.1% | 47.8% | **73.2%** |
| $\ell_\infty$ attacks | PGD-1000-$\ell_\infty$ | 0.0 % | 48.1% | 18.4% | 40.0% | 44.7% | 40.4% | 37.6% | 42.6% | 32.8% | 55.4% | **58.1%** |
| | FGSM | 9.2% | 55.2% | 30.4% | 46.7% | 54.3% | 39.9% | 42.7% | 46.4% | 40.1% | **62.4%** | 56.0% |
| | MI-FGSM | 0.0% | 53.9% | 0.9% | 30.2% | 50.8% | 37.8% | 31.6% | 45.3% | 31.2% | 60.1% | **72.9%** |
| | C&W-$\ell_\infty$ | 2.0% | 53.7% | 0.1% | 40.9% | 51.3% | 47.8% | 44.3% | 49.3% | 42.1% | 51.1% | **67.2%** |
| | SPSA | 0.4% | 50.2% | 0.6% | 19.4% | 45.8% | 33.2% | 30.4% | 44.9% | 38.4% | 68.2% | **69.8%** |
| | NATTACK | 1.9% | 50.1% | 2.1% | 19.7% | 43.1% | 34.3% | 30.9% | 45.1% | 40.2% | 45.4% | **64.2%** |
| | AutoAttack | 0.0% | 45.1% | 0.0% | 20.2% | 42.3% | 29.3% | 25.4% | 38.1% | 13.2% | 40.8% | **50.9%** |
| All attacks | - | 0.0% | 16.0% | 0.0% | 19.0% | 17.8% | 28.2% | 24.9% | 37.9% | 13.0% | 20.7% | **48.7%** |
| Clean accuracy | - | **89.7%** | 86.9% | 84.3% | 87.9% | 84.2% | 80.6% | 77.0% | 79.1% | 82.3% | 79.4% | 80.7% |

Table 6: Model robustness of ResNet-34 on Tiny-ImageNet over each individual attack (the higher the better).

| | | Vanilla | AVG | MAX | MSD | MN | MBN | **GBN** |
|---|---|---|---|---|---|---|---|---|
| $\ell_1$ attacks | PGD-$\ell_1$ | 10.0% | 28.1% | 17.1% | 9.0% | 12.4% | 45.2% | **55.7%** |
| | BBA | 5.8% | 27.5% | 26.2% | 7.6% | 8.8% | 37.0% | **45.2%** |
| $\ell_2$ attacks | PGD-$\ell_2$ | 12.3% | 30.5% | 24.4% | 14.2% | 18.1% | 31.7% | **53.9%** |
| | C&W-$\ell_2$ | 10.9% | 29.1% | 25.2% | 11.2% | 19.2% | 31.4% | **44.8%** |
| | Gaussian Noise | 54.2% | 41.6% | 36.3% | 26.7% | 40.2% | 46.2% | 42.8% |
| | BA | 27.1% | 35.2% | 24.9% | 13.7% | 30.2% | 38.4% | **44.1%** |
| $\ell_\infty$ attacks | PGD-$\ell_\infty$ | 7.1% | 6.9% | 4.7% | 7.6% | 19.8% | 40.2% | **50.2%** |
| | FGSM | 3.4% | 11.1% | 6.8% | 8.9% | 22.4% | 41.4% | **50.3%** |
| | MI-FGSM | 1.4% | 9.4% | 6.0% | 8.0% | 15.1% | 36.3% | **53.7%** |
| | SPSA | 0.3% | 7.1% | 6.5% | 10.0% | 29.8% | 39.2% | **52.0%** |
| | NATTACK | 0.8% | 8.2% | 7.5% | 12.3% | 29.9% | 40.3% | **48.5%** |
| | AutoAttack | 0.0% | 4.6% | 2.9% | 5.1% | 6.8% | 20.1% | **39.8%** |
| All attacks | - | 0.0% | 4.6% | 2.2% | 5.0% | 6.7% | 20.0% | **38.9%** |
| Clean accuracy | - | **54.2%** | 41.6% | 36.3% | 28.8% | 46.7% | 45.7% | 43.4% |

## E.5 Prediction Results of the gated sub-network

In this part, we show the prediction results of the gated sub-network in GBN, *i.e.*, the classification accuracy for the domain of different input samples. The results of LeNet on MNIST and ResNet-20 on CIFAR-10 are presented in Table 11. For MNIST, the gated sub-network of LeNet at different layers achieves high prediction accuracy for the input domains (*i.e.*, clean, $\ell_1$, $\ell_2$, and $\ell_\infty$ adversarial examples). However, on CIFAR-10, the prediction accuracy of the gated sub-network drops with the increasing of the layer depth. From the results, we can also confirm our findings in Section 4.3: (1) shallow layers are more critical to model robustness; and (2) as the layer depth increases, the features from different domains are highly entangled and mixed, making it harder for models to separate them. To further improve model robustness against multiple perturbations, we suggest to improve the prediction accuracy of the gated sub-network. We will further address it in the future studies.

Table 7: Model robustness of VGG-16 on CIFAR-10 over each individual attack (the higher the better).

| | | Vanilla | TRADES | $P_1$ | $P_2$ | $P_\infty$ | AVG | MAX | MSD | MN | MBN | **GBN** |
|---|---|---|---|---|---|---|---|---|---|---|---|---|
| $\ell_1$ attacks | PGD-$\ell_1$ | 32.8% | 27.9% | **72.2%** | 42.1% | 19.8% | 48.8% | 39.6% | 51.2% | 46.0% | 49.5% | 59.9% |
| | BBA | 0.0% | 18.2% | 24.8% | 36.1% | 18.7% | 45.9% | 35.5% | 44.9% | 41.1% | 46.8% | **69.5%** |
| $\ell_2$ attacks | PGD-$\ell_2$ | 32.9% | 60.9% | 21.7% | 64.4% | 62.8% | 61.4% | 60.0% | 64.9% | 50.4% | 62.1% | **70.7%** |
| | C&W-$\ell_2$ | 0.0% | 61.1% | 6.8% | 52.1% | 52.5% | 61.9% | 58.0% | 64.2% | 31.4% | 21.9% | **75.2%** |
| | Gaussian Noise | 82.0% | 79.9% | 70.1% | 86.0% | 82.8% | 74.9% | 62.9% | 69.9% | 61.8% | 51.2% | **79.9%** |
| | BA | 0.9% | 61.8% | 2.3% | 64.4% | 60.2% | 64.1% | 59.7% | 61.4% | 40.2% | 49.3% | **74.8%** |
| $\ell_\infty$ attacks | PGD-1000-$\ell_\infty$ | 0.0 % | 49.9% | 20.2% | 41.0% | 45.9% | 42.2% | 39.8% | 42.5% | 33.1% | 56.4% | **58.8%** |
| | FGSM | 9.1% | 56.2% | 32.1% | 47.7% | 55.8% | 41.0% | 44.6% | 47.3% | 41.8% | **63.7%** | 57.4% |
| | MI-FGSM | 0.3% | 54.2% | 1.2% | 31.8% | 51.9% | 39.3% | 33.4% | 47.0% | 32.7% | 60.9% | **73.8%** |
| | SPSA | 0.6% | 52.2% | 0.9% | 20.4% | 46.8% | 34.9% | 32.1% | 46.2% | 39.9% | 69.8% | **70.8%** |
| | NATTACK | 2.4% | 50.4% | 2.8% | 19.7% | 47.1% | 34.3% | 30.9% | 45.1% | 40.2% | 45.4% | **64.2%** |
| | AutoAttack | 0.0% | 46.1% | 0.0% | 20.2% | 43.3% | 30.4% | 27.4% | 39.2% | 14.3% | 40.8% | **51.9%** |
| All attacks | - | 0.0% | 17.9% | 0.0% | 19.5% | 18.5% | 30.2% | 27.0% | 38.9% | 14.1% | 21.5% | **50.6%** |
| Clean accuracy | - | **90.9%** | 87.1% | 84.9% | 88.1% | 85.0% | 82.1% | 79.2% | 78.4% | 80.2% | 84.3% | 84.1% |

Table 8: Model robustness of WideResNet-28-10 on CIFAR-10 over each individual attack (the higher the better).

| | | Vanilla | TRADES | $P_1$ | $P_2$ | $P_\infty$ | AVG | MAX | MSD | MN | MBN | **GBN** |
|---|---|---|---|---|---|---|---|---|---|---|---|---|
| $\ell_1$ attacks | PGD-$\ell_1$ | 24.7% | 27.2% | **71.4%** | 40.9% | 19.5% | 53.3% | 40.0% | 53.1% | 46.0% | 48.8% | 62.1% |
| | BBA | 0.0% | 16.8% | 23.1% | 35.6% | 19.1% | 49.0% | 36.1% | 45.2% | 42.0% | 47.6% | **71.8%** |
| $\ell_2$ attacks | PGD-$\ell_2$ | 27.1% | 60.8% | 21.8% | 63.7% | 61.2% | 65.3% | 62.6% | 63.1% | 51.1% | 61.9% | **70.9%** |
| | C&W-$\ell_2$ | 0.0% | 60.5% | 5.2% | 53.0% | 51.9% | 62.7% | 58.8% | 64.1% | 33.4% | 24.8% | **76.4%** |
| | Gaussian Noise | 91.2% | 80.2% | 69.9% | 85.9% | 82.6% | 78.8% | 70.3% | 77.8% | 63.9% | 54.2% | **77.8%** |
| | BA | 0.5% | 60.7% | 1.0% | 61.4% | 58.1% | 67.0% | 61.4% | 65.4% | 42.0% | 50.1% | **75.6%** |
| $\ell_\infty$ attacks | PGD-1000-$\ell_\infty$ | 0.4 % | 50.2% | 18.9% | 40.1% | 46.5% | 42.1% | 40.6% | 47.5% | 35.1% | 56.9% | **60.4%** |
| | FGSM | 8.7% | 58.2% | 32.4% | 47.4% | 52.6% | 39.9% | 44.4% | 52.2% | 42.0% | **65.2%** | 58.1% |
| | MI-FGSM | 0.0% | 54.2% | 0.6% | 30.6% | 48.3% | 37.8% | 36.0% | 50.3% | 35.1% | 61.9% | **70.7%** |
| | SPSA | 0.2% | 50.1% | 0.7% | 19.6% | 42.3% | 33.2% | 34.8% | 47.2% | 39.9% | 69.0% | **70.8%** |
| | NATTACK | 1.9% | 50.6% | 2.1% | 19.7% | 47.4% | 34.3% | 36.1% | 46.1% | 43.0% | 47.9% | **66.0%** |
| | AutoAttack | 0.0% | 47.2% | 0.0% | 20.0% | 44.8% | 31.3% | 30.4% | 40.2% | 15.8% | 38.8% | **52.4%** |
| All attacks | - | 0.0% | 16.5% | 0.0% | 19.0% | 18.8% | 31.3% | 30.9% | 40.0% | 15.6% | 24.5% | **51.5%** |
| Clean accuracy | - | **93.2%** | 87.6% | 83.4% | 88.5% | 85.1% | 83.2% | 79.1% | 80.3% | 83.6% | 84.1% | 83.9% |

### E.6 ALTERNATIVE PREDICTION APPROACH OF THE GATED SUB-NETWORK

In the main body of our paper, we calculate the normalized output using the gated sub-network in a soft-gated way (denoted "soft") based on Eqn. 3. In this section, we also try taking the top-1 prediction of $\mathbf{g}$ (the hard label) to normalize the output as an alternative approach (denoted "hard"). As shown in Table 12, the two approaches achieve comparable performance in terms of both robustness and standard accuracy.

### E.7 ATTACKS OUTSIDE THE PERTURBATION MODEL

In this section, we present some additional experiments exploring our model's performance on attacks which lying outside the perturbation model.

We conduct two experiments: (1) adversarially train a GBN model on $\ell_1$ and $\ell_\infty$ perturbations, and evaluate it on $\ell_2$ adversaries (denoted GBN$_{\ell_1+\ell_\infty}$); and (2) adversarially train a GBN model on $\ell_2$ and $\ell_\infty$ perturbations, and evaluate it on $\ell_1$ adversaries (denoted GBN$_{\ell_2+\ell_\infty}$). We use PGD-$\ell_1$, PGD-$\ell_2$, and PGD-$\ell_\infty$ attacks in Appendix D.1. The results are shown in Table 13.

Table 9: Robustness evaluation by AutoAttack (the higher the better). We report the clean accuracy, the robust accuracy of the individual attacks as well as the combined one of AutoAttack (denoted AA) using LeNet, ResNet-20, and ResNet-34.

| | Vanilla | TRADES | $P_1$ | $P_2$ | $P_\infty$ | AVG | MAX | MSD | MN | MBN | **GBN** |
|---|---|---|---|---|---|---|---|---|---|---|---|
| **MNIST** - $\ell_\infty$ | | | | | | | | | | | |
| $\text{APGD}_{CE}$ | 34.0% | 90.4% | - | - | - | 48.4% | 60.2% | 39.3% | 74.7% | 20.2% | 80.9% |
| $\text{APGD}_{DLR}$ | 0.0% | 82.1% | - | - | - | 41.2% | 55.7% | 33.9% | 0.0% | 52.2% | 88.4% |
| $\text{FAB}^T$ | 0.2% | 85.0% | - | - | - | 52.1% | 60.1% | 37.3% | 0.0% | 80.7% | 96.1% |
| Square | 0.0% | 81.8% | - | - | - | 50.2% | 53.1% | 38.5% | 0.0% | 59.7% | 79.3% |
| AA | 0.0% | 77.1% | - | - | - | 46.1% | 60.1% | 39.0% | 0.0% | 20.0% | 71.7% |
| **CIFAR-10** - $\ell_\infty$ | | | | | | | | | | | |
| $\text{APGD}_{CE}$ | 0.0% | 49.1% | 0.0% | 20.6% | 44.9% | 30.8% | 29.9% | 44.3% | 20.8% | 53.5% | 60.8% |
| $\text{APGD}_{DLR}$ | 0.0% | 50.2% | 0.0% | 18.8% | 43.2% | 28.1% | 25.7% | 39.1% | 29.9% | 62.4% | 58.9% |
| $\text{FAB}^T$ | 0.0% | 49.8% | 0.8% | 19.6% | 41.2% | 29.6% | 26.3% | 39.3% | 30.1% | 40.9% | 70.2% |
| Square | 0.1% | 47.4% | 4.6% | 28.4% | 39.0% | 40.5% | 31.6% | 46.1% | 17.8% | 14.5% | 60.3% |
| AA | 0.0% | 45.1% | 0.0% | 20.2% | 42.3% | 29.3% | 25.4% | 38.1% | 13.2% | 40.8% | 50.9% |
| **Tiny-ImageNet** - $\ell_\infty$ | | | | | | | | | | | |
| $\text{APGD}_{CE}$ | 0.0% | - | - | - | - | 5.7% | 3.7% | 6.3% | 8.7% | 39.3% | 44.3% |
| $\text{APGD}_{DLR}$ | 0.0% | - | - | - | - | 5.1% | 2.6% | 4.2% | 10.4% | 41.0% | 43.8% |
| $\text{FAB}^T$ | 0.2% | - | - | - | - | 5.2% | 2.7% | 4.3% | 9.1% | 37.3% | 50.0% |
| Square | 0.7% | - | - | - | - | 10.1% | 4.0% | 5.9% | 6.4% | 29.3% | 39.4% |
| AA | 0.0% | - | - | - | - | 4.6% | 2.9% | 5.1% | 6.8% | 20.1% | 39.8% |

Table 10: Model performance of ResNet-20 on CIFAR-10 on different PGD-$k$ attacks ($k$ denotes the iteration steps).

| | Vanilla | TRADES | AVG | MAX | MSD | MN | MBN | **GBN(ours)** |
|---|---|---|---|---|---|---|---|---|
| PGD-50 | 2.4% | 49.8% | 42.1% | 39.5% | 44.3% | 34.6% | 58.6% | 60.0% |
| PGD-100 | 0.0% | 49.5% | 41.7% | 39.2% | 44.1% | 34.1% | 58.2% | 59.6% |
| PGD-200 | 0.0% | 49.3% | 41.4% | 38.7% | 43.5% | 33.6% | 57.7% | 59.0% |
| PGD-1000 | 0.0% | 48.1% | 40.4% | 37.6% | 42.6% | 32.8% | 55.4% | 58.1% |

We empirically observe that training on 2 perturbation types actually improves overall robustness against all three perturbation adversaries. Unsurprisingly, the robustness of $\text{GBN}_{\ell_1+\ell_\infty}$ for $\ell_2$ adversarial examples and the robustness of $\text{GBN}_{\ell_2+\ell_\infty}$ for $\ell_1$ adversarial examples decrease to some degree, but the models still perform better than vanilla, and sometimes even outperform baseline methods.

In addition, we further evaluate model robustness on attacks outside the perturbation model using different prediction approaches of the gated sub-network (*i.e.*, soft label and hard label). As shown in Table 14, we achieve the similar results using the soft or hard label.

### E.8 DIFFERENT VARIANTS OF WHITE-BOX ATTACKS

Here, we examine the performance of GBN in more rigorous white-box attack scenarios. In this case, the adversary not only knows every detail of the GBN architecture, but also can manually select the BN branch to generate adversarial attacks.

We first generate adversarial attacks by fooling all the GBN layers in the model. Specifically, we generate adversarial attacks using PGD-$\ell_1$, PGD-$\ell_2$, and PGD-$\ell_\infty$ attacks in Appendix D.1 by making the gated sub-network to wrong predictions of the input domain. We force the generated perturbations to fool all the GBN layers in a ResNet-20 model at the same time. In particular, we first generate adversarial examples for different perturbation types using a ResNet-20 model; based on these examples for each domain, we then perturb them to fool the GBN layers. As shown in Table 15, our GBN still achieves high robustness against such adaptive attack. To further investigate this

Table 11: Gated sub-network prediction accuracy on inputs from different domains on different datasets (the higher the better). We demonstrate the prediction accuracy of gated sub-network at each layer from top to bottom. The adversarial examples are generated by PGD-$\ell_1$, PGD-$\ell_2$, and PGD-$\ell_\infty$ attacks in Appendix D.1.

(a) LeNet on MNIST

|  | Clean | $\ell_1$ | $\ell_2$ | $\ell_\infty$ |
|---|---|---|---|---|
| gate$_1$ | 96.7% | 78.2% | 92.3% | 98.6% |
| gate$_2$ | 94.2% | 74.5% | 89.9% | 98.8% |

(b) ResNet-20 on CIFAR-10

|  | Clean | $\ell_1$ | $\ell_2$ | $\ell_\infty$ |
|---|---|---|---|---|
| gate$_1$ | 79.7% | 85.2% | 88.7% | 84.8% |
| gate$_2$ | 77.2% | 84.5% | 85.9% | 83.7% |
| gate$_3$ | 77.4% | 84.2% | 82.3% | 83.9% |
| gate$_4$ | 75.2% | 83.5% | 75.2% | 82.1% |
| gate$_5$ | 74.7% | 82.2% | 73.4% | 81.6% |
| gate$_6$ | 73.2% | 82.5% | 69.1% | 79.8% |
| gate$_7$ | 71.7% | 81.2% | 67.9% | 78.3% |
| gate$_8$ | 65.2% | 74.5% | 67.2% | 75.7% |
| gate$_9$ | 64.7% | 74.2% | 66.8% | 75.5% |
| gate$_{10}$ | 64.2% | 73.6% | 65.7% | 74.8% |
| gate$_{11}$ | 64.7% | 73.2% | 64.6% | 74.5% |
| gate$_{12}$ | 64.2% | 72.9% | 64.4% | 73.9% |
| gate$_{13}$ | 63.9% | 71.2% | 64.1% | 73.7% |
| gate$_{14}$ | 63.0% | 69.5% | 64.0% | 72.3% |
| gate$_{15}$ | 64.1% | 69.1% | 64.2% | 71.6% |
| gate$_{16}$ | 64.2% | 68.7% | 63.9% | 70.9% |
| gate$_{17}$ | 62.5% | 68.2% | 63.9% | 70.6% |
| gate$_{18}$ | 63.2% | 68.5% | 63.6% | 69.5% |
| gate$_{19}$ | 62.7% | 67.9% | 63.4% | 69.4% |

Table 12: Model robustness of ResNet-20 on CIFAR-10 using different prediction approaches of the gated sub-network (the higher the better). We use PGD-$\ell_1$, PGD-$\ell_2$, and PGD-$\ell_\infty$ attacks in Appendix D.1.

|  | vanilla | hard | soft |
|---|---|---|---|
| PGD-$\ell_1$ | 31.4% | 58.2% | 58.5% |
| PGD-$\ell_2$ | 32.1% | 69.6% | 69.5% |
| PGD-$\ell_\infty$ | 3.2% | 59.9% | 60.1% |
| Clean accuracy | 89.7% | 80.4% | 80.7% |

Table 13: Attacks outside the perturbation model of ResNet-20 on CIFAR-10 (the higher the better).

|  | Vanilla | AVG | MAX | MSD | GBN | GBN$_{\ell_1+\ell_\infty}$ | GBN$_{\ell_2+\ell_\infty}$ |
|---|---|---|---|---|---|---|---|
| PGD-$\ell_1$ | 31.7% | 47.4% | 38.1% | 49.9% | 58.5% | 57.4% | 40.9% |
| PGD-$\ell_2$ | 32.1% | 59.2% | 58.0% | 62.8% | 69.5% | 41.8% | 65.4% |
| PGD-$\ell_\infty$ | 3.2% | 42.1% | 39.7% | 44.5% | 60.1% | 57.2% | 56.8% |
| Clean accuracy | 89.7% | 80.6% | 77.0% | 79.1% | 80.7% | 81.3% | 81.1% |

Table 14: Attacks outside the perturbation model of ResNet-20 on CIFAR-10 using different prediction approaches of the gated sub-network ("h" denotes the hard label and "s" denotes the soft label).

| | $GBN_{\ell_1+\ell_\infty}(s)$ | $GBN_{\ell_2+\ell_\infty}(s)$ | $GBN_{\ell_1+\ell_\infty}(h)$ | $GBN_{\ell_2+\ell_\infty}(h)$ |
|---|---|---|---|---|
| PGD-$\ell_1$ | 57.4% | 40.9% | 57.9% | 41.5% |
| PGD-$\ell_2$ | 41.8% | 65.4% | 40.2% | 67.9% |
| PGD-$\ell_\infty$ | 57.2% | 56.8% | 58.2% | 56.1% |
| Clean accuracy | 81.3% | 81.1% | 81.2% | 80.9% |

phenomena, we also provide the prediction accuracy of the gated sub-network in GBN at different layers given adversarial examples above. As can be found in Table 16: it is hard to fool all GBN layers within a model. In other words, though some GBN layers are fooled to misclassify the domains of the samples, some other GBN layers could still keep comparatively high prediction accuracy.

We then generate adversarial examples by manually selecting the BN branches. Specifically, we generate $\ell_1$, $\ell_2$, and $\ell_\infty$ adversarial examples through $BN^0$, $BN^1$, $BN^2$, and $BN^3$ in GBN, respectively. The hyper-parameters for PGD-$\ell_1$, PGD-$\ell_2$, and PGD-$\ell_\infty$ attacks in Appendix D.1. Note that their statistics are estimated by clean, $\ell_1$, $\ell_2$, and $\ell_\infty$ adversarial examples during training, respectively. As presented in Table 17, the model achieves good robustness on different datasets in this more rigorous white-box attack scenario. In particular, the worst-case accuracies among all branches are largely comparable to the results presented in Table 1, suggesting that the robustness of GBN does not rely on obfuscated gradients (Athalye et al., 2018).

Table 15: White-box attacks on CIFAR-10 using ResNet-20 by fooling all the GBN layers.

| $BN^0$ | $BN^1$ | $BN^2$ | $BN^3$ |
|---|---|---|---|
| 73.2% | 71.4% | 72.4% | 71.8% |

### E.9 VISUALIZATION OF FEATURES

Here, we further provide the visualization of features before and after the normalization blocks of each method using t-SNE. Specifically, given 100 CIFAR-10 clean samples from a specific class (*e.g.*, "dog"), we first generate corresponding $\ell_1$, $\ell_2$, and $\ell_\infty$ adversarial examples; we then visualize the features before and after the normalization blocks (*i.e.*, the BN block for AVG and the GBN block for our method). Note that we use the hyper-parameters of PGD-$\ell_1$, PGD-$\ell_2$, and PGD-$\ell_\infty$ attacks in Appendix D.1. As can be seen in Figure 6, the features of different domains are aligned to domain-invariant representations after normalized by GBN. Specifically, after the normalization of GBN, one clean example (the red dot) and its corresponding adversarial examples (the corresponding blue, green, and yellow dots) are more likely to be close in the space of normalized output (Figure 6(b)). However, after the normalization of BN (Figure 7), the distance between the clean example and its corresponding adversarial examples are still far, and we cannot figure our any tendency between Figure 7(a) and Figure 7(b).

### E.10 MORE DETAILS OF THE RUNNING STATISTICS OF DIFFERENT MODELS

We train a VGG-16 and WideResNet-28-10 model with the multiple BN branch structure and inspect the running statistics of each BN at different layers, as shown in Figure 8 and Figure 9. Surprisingly, the running statistics between different domains in the first layer are almost indistinguishable. However, according to the ablation study in , adding GBN into shallow layers is the most robust than adding GBN to other layers. We may understand it from three following viewpoints: (1) gated sub-network prediction accuracy. We provided the prediction accuracy of the gated sub-network on inputs from different domains in Table 11. According to the experimental results, the gated sub-network achieves the highest prediction accuracy in the first layer, though the running statistics in

Table 16: Gated sub-network prediction accuracy on adversarial examples generated to fool the GBN layers. We demonstrate the prediction accuracy of gated sub-network at each layer from top to bottom.

|  | Clean | $\ell_1$ | $\ell_2$ | $\ell_\infty$ |
|---|---|---|---|---|
| $\text{gate}_1$ | 69.7% | 48.2% | 54.7% | 70.2% |
| $\text{gate}_2$ | 70.2% | 50.3% | 58.2% | 73.3% |
| $\text{gate}_3$ | 70.4% | 51.4% | 57.3% | 72.1% |
| $\text{gate}_4$ | 71.1% | 50.1% | 56.4% | 74.0% |
| $\text{gate}_5$ | 68.2% | 50.9% | 53.1% | 73.6% |
| $\text{gate}_6$ | 67.4% | 49.0% | 50.2% | 73.5% |
| $\text{gate}_7$ | 65.1% | 49.2% | 51.0% | 72.4% |
| $\text{gate}_8$ | 60.3% | 48.3% | 49.2% | 70.5% |
| $\text{gate}_9$ | 60.8% | 46.1% | 49.3% | 69.8% |
| $\text{gate}_{10}$ | 60.5% | 44.0% | 48.2% | 69.1% |
| $\text{gate}_{11}$ | 62.2% | 42.1% | 46.2% | 69.4% |
| $\text{gate}_{12}$ | 60.2% | 41.1% | 46.3% | 67.3% |
| $\text{gate}_{13}$ | 61.9% | 40.3% | 45.1% | 66.8% |
| $\text{gate}_{14}$ | 58.1% | 40.4% | 44.2% | 66.9% |
| $\text{gate}_{15}$ | 59.2% | 40.2% | 44.4% | 65.4% |
| $\text{gate}_{16}$ | 59.3% | 41.3% | 41.5% | 63.2% |
| $\text{gate}_{17}$ | 54.4% | 41.4% | 42.9% | 60.1% |
| $\text{gate}_{18}$ | 54.1% | 40.4% | 41.4% | 60.5% |
| $\text{gate}_{19}$ | 54.2% | 40.0% | 40.3% | 60.0% |

Table 17: White-box attacks on MNIST and CIFAR-10 by manually selecting BN branches to generate adversarial examples ($\text{BN}^0$, $\text{BN}^1$, $\text{BN}^2$, and $\text{BN}^3$ denotes the BN branch for clean, $\ell_1$, $\ell_2$, and $\ell_\infty$ adversarial examples in GBN, respectively).

(a) LeNet on MNIST

|  | $\text{BN}^0$ | $\text{BN}^1$ | $\text{BN}^2$ | $\text{BN}^3$ |
|---|---|---|---|---|
| PGD-$\ell_1$ | 86.8% | 88.5% | 86.9% | 97.1% |
| PGD-$\ell_2$ | 97.8% | 98.9% | 97.8% | 98.7% |
| PGD-$\ell_\infty$ | 97.1% | 98.6% | 96.3% | 95.4% |
| Clean accuracy | 98.7% | 98.7% | 98.7% | 98.7% |

(b) ResNet-20 on CIFAR-10

|  | $\text{BN}^0$ | $\text{BN}^1$ | $\text{BN}^2$ | $\text{BN}^3$ |
|---|---|---|---|---|
| PGD-$\ell_1$ | 58.5% | 56.4% | 59.9% | 60.3% |
| PGD-$\ell_2$ | 69.7% | 69.4% | 68.2% | 69.5% |
| PGD-$\ell_\infty$ | 60.3% | 61.1% | 61.3% | 59.2% |
| Clean accuracy | 80.7% | 80.7% | 80.7% | 80.7% |

the first layer are almost indistinguishable. (2) the convolutional layers in the gated sub-network may still be able to distinguish the differences between these features (see discussions in Appendix C). (3) adding GBN to the first layer shows better performance than solely adding GBN to deeper layers might because those shallow layers are more critical to model robustness Liu et al. (2019b). We will study it in future work.

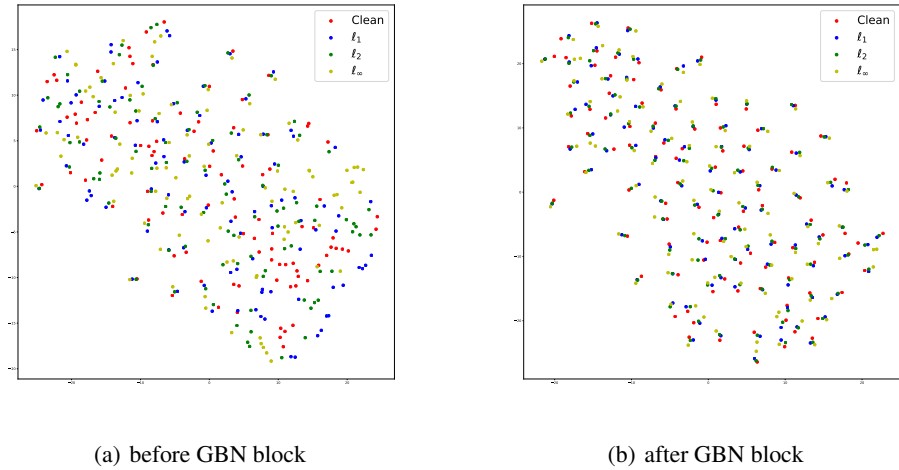

(a) before GBN block                         (b) after GBN block

Figure 6: We use the GBN block at layer $conv3\_1$ of a VGG-16 model. (a) demonstrates the visualization of features before the GBN block; (b) shows the visualization of features after the GBN block.

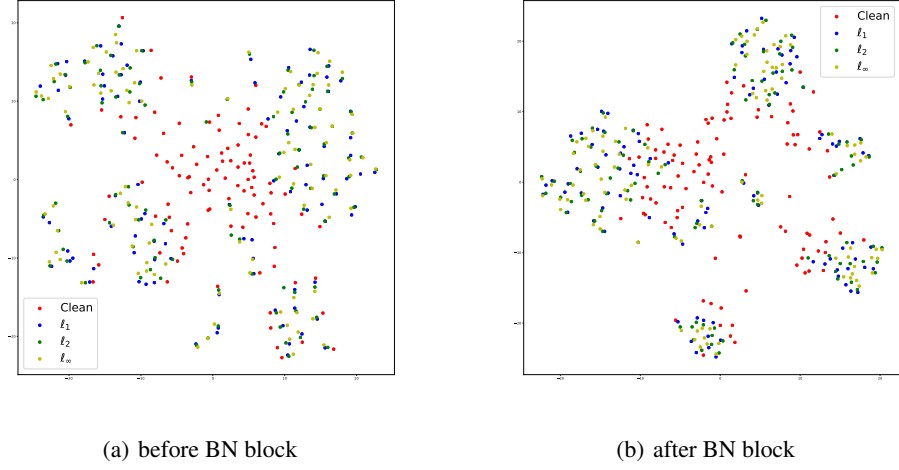

(a) before BN block                           (b) after BN block

Figure 7: We use the BN block at layer $conv3\_1$ of a VGG-16 model trained using AVG. (a) demonstrates the visualization of features before the BN block; (b) shows the visualization of features after the BN block.

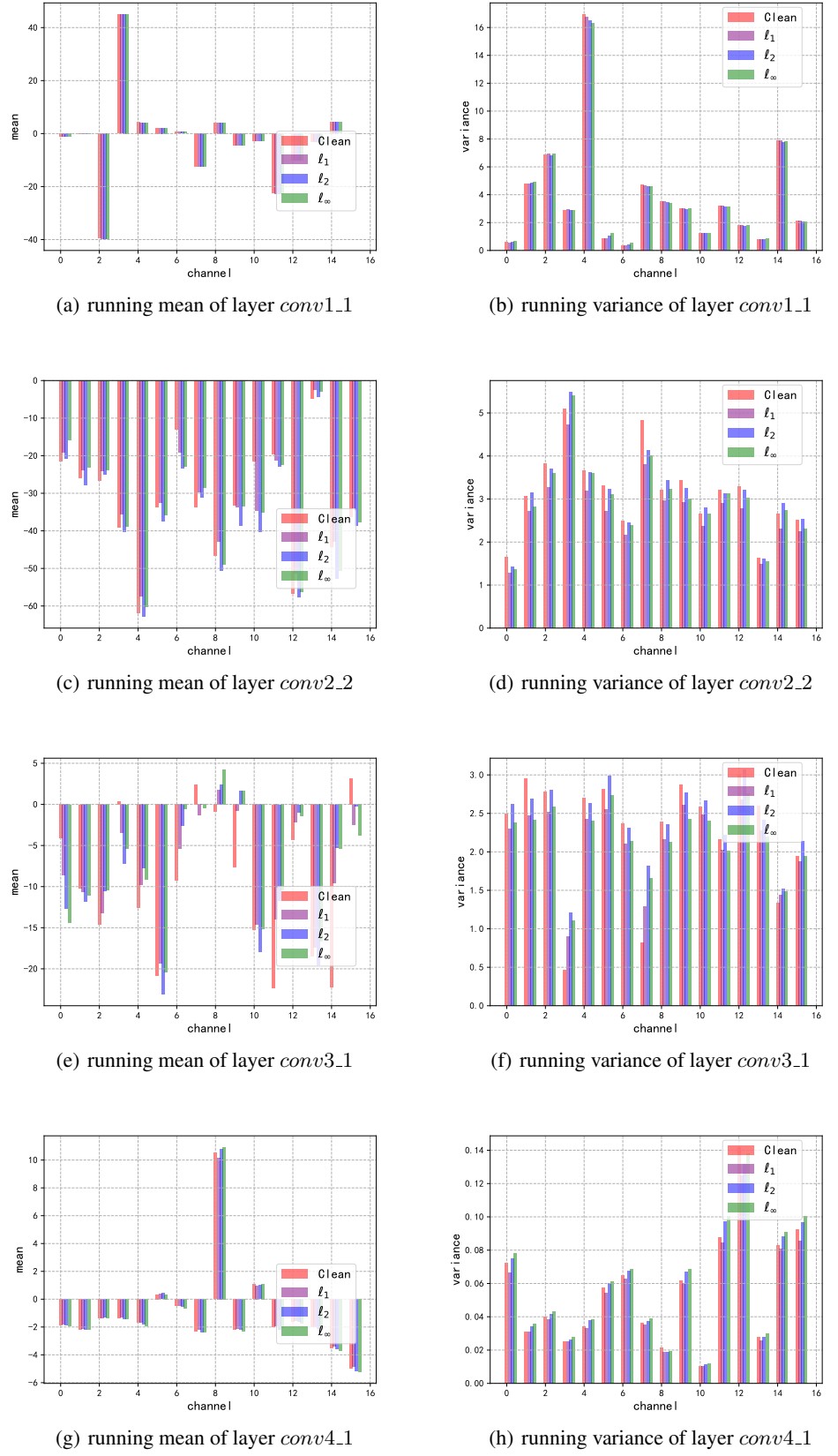

(a) running mean of layer $conv1\_1$

(b) running variance of layer $conv1\_1$

(c) running mean of layer $conv2\_2$

(d) running variance of layer $conv2\_2$

(e) running mean of layer $conv3\_1$

(f) running variance of layer $conv3\_1$

(g) running mean of layer $conv4\_1$

(h) running variance of layer $conv4\_1$

Figure 8: Running statistics (running mean and running variance) of each BN in the multiple BN branches at different layers of a VGG-16 model on CIFAR-10.

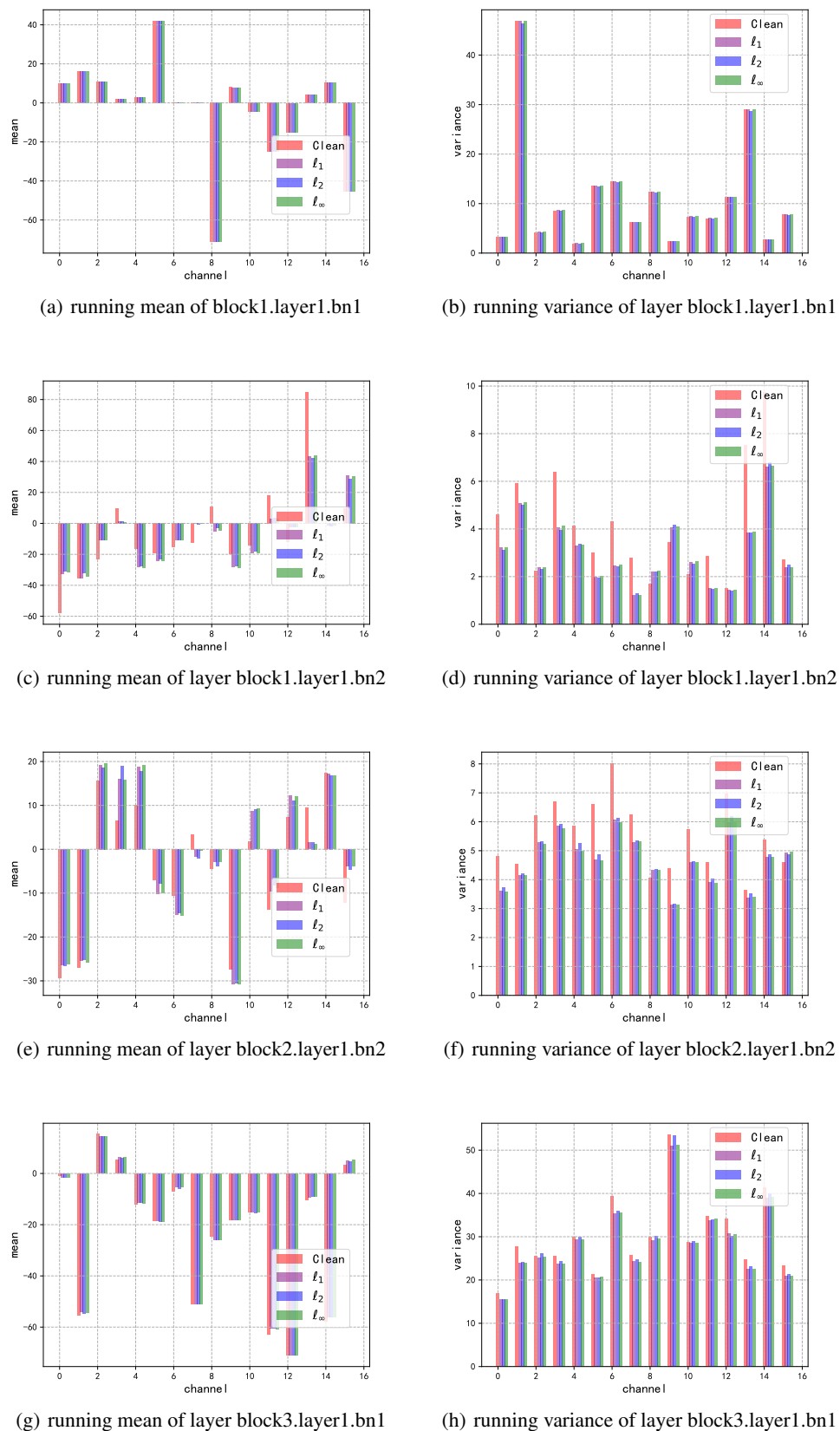

(a) running mean of block1.layer1.bn1

(b) running variance of layer block1.layer1.bn1

(c) running mean of layer block1.layer1.bn2

(d) running variance of layer block1.layer1.bn2

(e) running mean of layer block2.layer1.bn2

(f) running variance of layer block2.layer1.bn2

(g) running mean of layer block3.layer1.bn1

(h) running variance of layer block3.layer1.bn1

Figure 9: Running statistics (running mean and running variance) of each BN in the multiple BN branches at different layers of a WideResNet-28-10 model on CIFAR-10.

