# OpenReview forum: "Towards Defending Multiple Adversarial Perturbations via Gated Batch Normalization"
_ICLR.cc/2021/Conference — Reject_

### Official Review · AnonReviewer2 · 2020-10-27
**Interesting results, novelty slightly limited – would benefit from further exploration**

**Rating:** 6
**Confidence:** 4

**Review:**

### Summary

The authors propose to improve the robustness against adversarial attacks in deep networks via the use of a customized normalization strategy. Their central methodological suggestion is called gated batch normalization (GBN), and involves a (soft, but some other variants are briefly explored as well) gating mechanism that maps to distinct normalization branches. Their idea is evaluated on a set of benchmarks and robustness is measured threefold: in terms of $L_1$, $L_2$ and $L_\infty$ perturbations.

### Strengths

The authors formulate an intellectually compelling underlying hypothesis, namely that adversarial perturbations can be viewed as coming from distinct domains, special in character since they may be positioned very closely to the original domain of natural images of the unperturbed distribution. This is an interesting viewpoint (although I am not sure why we would want to call these entities domains, a "mode" or "manifold" would seem more appropriate), and this aspect is in my opinion the strong point of the paper!

The experimental section evaluates GBN against competitor approaches, either directly purposed for adversarial robustness (e.g. MSD) and conceptually similar normalization techniques (MN, MBN). CGN achieves convincing performance, and the underlying idea of separating BN statistics seems to improve adversarial robustness in a substantial way.

### Weaknesses

While the methodology of several gated BN units is well motivated, the technical novelty is somewhat limited. In particular, the method proposed here is very similar to the work of Deecke et al. (ICLR, 2019), with its main differentiation being that the gating mechanism is computed in a round-robin fashion (i.e. parameters are fixed when updating the gates, and vice versa) and with ground-truth information passed to the normalization units.

That being said, introducing supervision (with respect to the initially applied perturbation) seems to be crucial to be effective when defending against adversarial examples, and increases performance against a range of competing approaches. Because of this, I would argue that the limited novelty in terms of the normalization is offset somewhat by the idea to try this out in an adversarial setting – and getting it to work there.

### Suggestions

There are a few suggestions that I would have liked to see included in the manuscript to increase its strength:
* further study around what constitutes a good gating mechanism, in particular (i) what is the performance for a Gumbel-based approach; (ii) since the gates are computed offline, is there some sort of non-local parameter-sharing that (e.g. via a regularizer) benefits performance? This would also help in increasing the novelty of this work respective to the various competitive normalization approaches introduced in earlier publications.
* Additional analysis around Figure 1 (c) and (d). Why was this particular layer chosen? Does this trend hold in other layers as well? What about deep lower versus higher layers. There are a number of interesting questions to be explored around this theme, for example is this trend a function of the semanticity? For some background reading on the depth of networks and their semantic function, I recommend looking into Asano et al. (ICLR, 2020).

### Minor suggestions

Good job on Figure 1 (a) and (b), this helped tremendously with quickly understanding the proposed mechanism! However, Figure 2 could use some improvement (in fact, maybe this can be removed in favor of additional discussions as outlined above).

There seems to be some issue with the citations, e.g. Kurakin et al. (ICLR, 2017) and Deecke et al. (ICLR, 2019) are cited with first & last names in the wrong order.

---

> ### Author Response · Authors · 2020-11-16
> **Clarify the novelty and add more explorations**
>
> We thank the reviewer for the positive supports and constructive suggestions in this COVID-19 outbreak! Stay safe and healthy! We will clarify our novelty and are happy to follow your suggestions to add additional discussions in the revision soon. We hope the response and revision can be helpful to address your concerns and increase our strength.
>
> ***Q1***: Novelty compared to Deecke et al. (ICLR, 2019) (MN):
>
> ***A***: Though looking similar, our GBN is not a simple modification or the introduction of MN into another scenario. Otherwise, the performance of MN against adversarial attacks should be great (however, MN performs far worse than GBN in Table 1 a,b,c by large margins).
>
> This paper indeed conveys a general idea to defend multiple adversarial perturbation types: align the distribution among different perturbations by using one normalization module to handle one perturbation type (learn domain-specific features), and then the aligned distribution contribute to the subsequent modules or layers (e.g., convolutional or fully connected layer) for learning domain-invariant representations. Our design and training strategy of GBN ensure that the normalized output for each perturbation type has a Gaussian distribution, and these Gaussian distributions among different perturbations are well aligned (e.g., one clean example $\mathbf{x}$ and its corresponding adversarial examples $\mathbf{x}+\delta^1$, $\mathbf{x}+ \delta^2$, etc. are very close in the space of normalized output). By doing this, the other modules  (e.g., convolutional or fully connected layer) are easier to learn the native representation of the data (domain-invariant representations).
>
> We argue that MN cannot align the distributions. On the contrary, MN aims to preserve the diversity among different distributions, by learning the mixture of Gaussian, not one single Gaussian. This idea of MN may work well for the multi-task learning scenario, where the data distributions for different tasks are significantly different, by avoiding normalizing the data in a Gaussian (like the naive BN does). For example, in their original paper, MN works well when classifying the data mixed with MNIST and CIFAR-10 datasets for a 20 classes problem (10 classes of MNIST + 10 classes of CIFAR-10), which is significantly different from the problem of defending multiple adversarial perturbation types (e.g., we classify the data mixed with clean and its corresponding adversarial examples in MNIST for a 10 classes problem).
>
> Here, we highlight that it is essential to defend multiple adversarial perturbation types by aligning the distributions among different domains, which is also an essential idea in domain adaptation. Moreover, we use 'domain', rather than 'mode' or 'manifold', for inheriting the description from domain adaptation.
>
> We will add more discussions and comparisons to MN in our revision soon.
>
> ***Q2***: Further studies of the gating mechanism:
>
> ***A***: Thanks! These are definitely interesting directions, and we will explore them in our future work.
>
> ***Q3***: Additional studies around Figure 1 (c) and (d):
>
> ***A***: In Figure 1, we randomly picked up a layer due to the space limitation in the main text. We provided more discussion for different layers in Section 4.3, and the detailed results of running statistics for different layers were shown in Appendix E.8. We will make these points clearer in our revision.
> Moreover, inspired by Asano et al. (ICLR, 2020), we further conjecture the reasons for the phenomenon mentioned in Figure 3 and 6. Since shallow layers contain limited information about the image statistics and can be learned via a few of samples, it is more simple to separate features in the shallow layers compared to the deep layers. However, the reason for the indistinguishability of features in the first layer is still unexplored, which we leave for future works. We will cite the paper and add related discussions in our revision.
>
> ***Q4***: Figure 2 could use some improvement:
>
> ***A***: Thanks! We will cut the space in Figure 2 and add more discussions mentioned above.
>
> ***Q5***: Citation issues:
>
> ***A***: Thanks! We will solve the citation problem.

---

### Official Review · AnonReviewer1 · 2020-10-27
**Method is interesting. The results show the effectiveness. But the reviewer worries whether the evaluation is valid.**

**Rating:** 5
**Confidence:** 4

**Review:**

Existing defenses are usually robust against certain types of attacks. This paper proposes to build a defense against different types of norm-bounded attacks via gated batch normalization. This work treats different types of adversarial attacks as data from different domains and proposes to use gated batch normalization to improve the robustness across different domains. The idea is interesting. The result shows the effectiveness of the defense.

Weakness and Concerns:

1. Adaptive attack for the GBN layer is missing. For example, one possible attack is to attack the GBN layer directly. Say the L_inf attack's aim is to fool the GBN layer, such that the gate output is misled from dealing with the L_inf but the L_0, the defense should fail. The reviewer worries that the robustness is not measured correctly, maybe rely on the obfuscated gradient [5].

2. No comparison to the STOA method. For example, the L_inf on LeNet MNIST does not compare to the TRADES [2] paper. On CIFAR-10, the author should also provide results on the WRN model [3] given the rich literature using it. Also, the standard CW and PGD attack for L_inf attack is not provided here. The paper should also compare to [4], as it also builds robustness against different kinds of norm bounded attacks.

3. Can the author list exactly how many steps they used for each attack evaluation? Does the paper use the best attack algorithm for each norm bounded attack? For example, it is known that CW/PGD is effective for L_inf attack.

4. Theory only considers the linear case.

5. Can the author compare to baseline model TRADES [2]? Since it can also be used to align the features from different types of attacks. Without such comparison, it is not clear how much gain the proposed method achieves compared with the established benchmark.

6. What is the difference between this method and [1]?



[1] Xie et. al. Adversarial Examples Improve Image Recognition.
[2] Zhang et. al. Theoretically Principled Trade-off between Robustness and Accuracy.
[3] Madry et. al. Towards Deep Learning Models Resistant to Adversarial Attacks.
[4] Schott et. al. Towards the first adversarially robust neural network model on MNIST.
[5] Athalye et. al. Obfuscated Gradients Give a False Sense of Security: Circumventing Defenses to Adversarial Examples.

The rating will be updated based on the rebuttal.

---

> ### Author Response · Authors · 2020-11-16
> **Clarify the evaluation details and add more experiments**
>
> We thank the reviewer for appreciating our efforts and providing constructive comments in this COVID-19 outbreak! Stay safe and healthy. We will address all concerns and update a new revision soon, so we humbly expect you can reconsider the final decision.
>
> ***Q1***: Adaptive attack for the GBN layer is missing:
>
> ***A***: To prove that no obfuscated gradients have been introduced, we evaluated our GBN model on different SOTA black-box or gradient-free attacks (e.g., BA, SPSA, NAttack, and Square Attack). We then performed adaptive white-box attacks (see Appendix E.7), in which the adversary knows every detail of the model and manually select BN branches to generate adversarial attacks. Our model achieved strong robustness in these scenarios (see Table 4, 5, 6, 7, and12). Though there is no principled reason why adversaries could manually manipulate the model, we're happy to follow your suggestions and intentionally fool the GBN layer. We will add the adaptive attack for the GBN layer and provide more discussions in the revision soon.
>
> ***Q2 & Q5***: No comparison to the STOA method. Can the author compare to baseline model TRADES [2]?:
>
> ***A***: As for the compared method, since our goal is to defend multiple adversarial perturbations, we primarily compared with those SOTA baselines for improving model robustness against multiple perturbation types (e.g., AVG, MSD, etc.). As for the standard CW (CW for $\ell_2$ attack) and PGD attack for $\ell_{\infty}$ attack, we have already used them as our attacks in the last revision as C&W and PGD-$\ell_{\infty}$. As for Schott et al. [4], we already used it as one of our baseline which is denoted as ABS. We would also like to follow your suggestions and add the following contents in the revision soon: (a) compare with TRADES; (b) add experiments using WRN.
>
> ***Q3***: Can the author list exactly how many steps they used for each attack evaluation?:
>
> ***A***: We already provided all details for all attacks we used in Appendix D.1, e.g., the iteration step for PGD $\ell_1$, $\ell_2$, and $\ell_{\infty}$ is 50. We adopt the most widely used attacks for each norm-bounded attack following (Tramer & Boneh, 2019; Pratyush et al., 2020).  We would also like to make them clearer and add CW for $\ell_{\infty}$ attacks in our revision soon.
>
> ***Q4***: Theory only considers the linear case:
>
> ***A***: Thanks. Our objective here is to provide theoretical evidence to support our multi-domain hypothesis, and we take linear classifiers as such an example. For general classifiers, it is usually intractable to measure the distance between these different types of adversarial perturbation precisely. We're also happy to follow your suggestions and cut the space of the theorem in the main text in our revision soon.
>
> ***Q6***: What is the difference between this method and [1]?:
>
> ***A***: We already compared our GBN with [1] in Section 2.2. Briefly, we differ it in two parts: (a) [1] manually selects the BN branch for each input which requires prior knowledge, while GBN predicts the domain for inputs and then jointly incorporate the outputs of multiple BN branches to provide the final result; (b) [1] tries to improve image recognition, while we aim to defend multiple adversarial perturbations.

---

> > ### Comment · AnonReviewer1 · 2020-11-20
> > **The intuition for why Gated BN works is not clear, experimental results are not convincing and not consistent.**
> >
> > The reviewer thank the authors for the detailed feedback.
> >
> > 1. Adaptive attacks: I see the results in Appendix E.7, however, it does not convince me. First, could the author clarify why Table 12 results are not consistent across different types of attack? e.g.,  given L_inf attack, the model is most vulnerable when a white-box attack is attacking BN^3(L_inf BN), however, Given L_1 attack, the model is most ROBUST when attacking L_1BN  branch,   why would the white-box attack is not the most successful when trying to fool BN normalize towards the wrong BN gate, why are they not consistent? Second, on CIFAR-10, the robust performance on L_inf is over 59%, with a weak Res20 model, however, this is not consistent (too high) with the established work. (see [2, 3]). I published two papers on this benchmark, but I do not believe the number here is correct (too high, even higher than results in Table 1 without adaptive attack). Can the author clarify their exact setup? E.g., the number of steps for the attack.
> >
> >
> > 2. Could the author point out where they add CW and PGD attacks? Also, update comparison with the STOA method.
> >
> > 3. 50 steps are too few for measuring robustness, especially for a defense that potentially relies on gradient masking. Could the author try 1000 steps and see how much it drops (for example on CIFAR-10, L_inf, with standard PGD attack set up in [3])? I guess the accuracy will continuously drop. The author should report the number when accuracy converges. Again, I do not believe the result in Table 1 (b), such as L_inf robustness is that high on Res20, the attack may be too weak.
> >
> > 3. It seems L_0 attack is missing, is there any reason that the paper does not study it?
> >
> > 4. Can the author clarify the intuition for why this method should work?
> >
> > My other questions are addressed.

---

> > > ### Author Response · Authors · 2020-11-23
> > > **Clarify the experimental setting and results**
> > >
> > > We thank the reviewer for their constructive feedback. We address your concerns as follows:
> > >
> > > ***Q1***: Adaptive attacks.
> > >
> > > ***A***: We will clarify your questions as listed below:
> > >
> > > (a) According to our experimental results in Table 12, attacking the $\ell_1$ BN branch is not always the most robust for the $\ell_1$ attack (e.g., ResNet-20 is most vulnerable to  $\ell_1$ PGD attack generated by the $\ell_1$ BN branch on CIFAR-10). Our GBN normalize the output in a soft-gated way using the weighted average of multiple BN branches, instead of a single BN branch with the largest predicted weight. Thus, attacking the branch corresponding to the attack type is not necessarily the best.
> > >
> > > (b) As for the experimental results in Table 12, we used the Foolbox as implementations and adopted the same hyper-parameters for the PGD attacks with (Pratyush et al., 2020). As for TRADES and PGD adversarial training on single perturbation types [2,3], we already compared them and the results are consistent with the original paper. We will add them in the revision soon.
> > >
> > > We already evaluated white-box attacks in our main paper and reported the worst-case results using different BN branches as shown in Table 1. As you can see, the lowest accuracy among different branches for each perturbation type in Table 12 is mostly consistent with the model accuracy of the white-box attacks in Tabel 4, 5, 6. Thus, our white-box experiment in Table 1, 4, 5, 6 can be regarded as the worst-case results for our defense. Our intuition for the experiment in Appendix E.7 was to provide further and deeper analysis in such a more rigorous scenario specifically designed for our defense (multi-BN structure).
> > > As for the setup of attacks in Appendix E.7, we use the same setting of PGD attacks in the main experiment (c.f. Appendix D.1).  For PGD-$\ell_1$, on MNIST, we set the perturbation magnitude $\epsilon$=10, iteration number $k$=50, and step size $\alpha$=$\epsilon$/10; on CIFAR-10, we set the perturbation magnitude $\epsilon$=12, iteration number $k$=50, and step size $\alpha$=0.05. For PGD-$\ell_2$, on MNIST, we set the perturbation magnitude $\epsilon$=2.0, iteration number $k$=100, and step size $\alpha$=0.1; on CIFAR-10, we set the perturbation magnitude $\epsilon$=0.5, iteration number $k$=50, and step size $\alpha$=$\epsilon$/10. For PGD-$\ell_{\infty}$, on MNIST, we set the perturbation magnitude $\epsilon$=0.3, iteration number $k$=50, and step size $\alpha$=0.01; on CIFAR-10, we set the perturbation magnitude $\epsilon$=0.03, iteration number $k$=40, and step size $\alpha$=$\epsilon$/10. We will make it clearer in the revision soon.
> > >
> > >
> > > ***Q2***: CW and PGD attacks.
> > >
> > > ***A***: We already provided the C&W for $\ell_2$ attacks and PGD for $\ell_1$, $\ell_2$, $\ell_{\infty}$ attacks in our initial submission in Appendix D.1. The results for these attacks were shown in Table 4, 5, and 6. We also follow your suggestions and add C&W for $\ell_{\infty}$ attacks and compare with TRADES in our revision.
> > >
> > > ***Q3***: 50 steps are too few for measuring robustness, especially for a defense that potentially relies on gradient masking.
> > >
> > > ***A***: We add additional PGD attacks with 50, 100, 200, 1000 steps for on CIFAR-10 in terms of $\ell_{\infty}$ norms in our revision. All explanations of why our approach should be better could be deferred for Q5.
> > >
> > > ***Q4***: L_0 attack is missing
> > >
> > > ***A***: In our paper, we do not use $\ell_0$ attack for two main reasons: (a) following (Tramer & Boneh, 2019; Pratyush et al., 2020; Croce & Hein, 2020a), we evaluate model robustness against $ell_1$, $ell_2$, and $ell_{\infty}$ attacks, which are the most representative and commonly used adversarial perturbation types in the multiple perturbations literature; (b) $\ell_1$ adversary with radius $\epsilon$ is strictly stronger than an $\ell_0$ adversary with the same radius, and so we choose to explicitly defend against $\ell_1$ perturbations instead of the $\ell_0$ perturbations (Pratyush et al., 2020).
> > >
> > > ***Q5***: clarify the intuition for why this method should work
> > >
> > > ***A***: We believe the intuition of our method can be summarized as: align the distribution among different perturbations by using one normalization module to handle one perturbation type (learn domain-specific features), and then the aligned distribution contribute to the subsequent modules or layers (e.g., convolutional or fully connected layer) for learning domain-invariant representations. Our design and training strategy of GBN ensure that the normalized output for each perturbation type has a Gaussian distribution, and these Gaussian distributions among different perturbations are well aligned (e.g., one clean example $\mathbf{x}$ and its corresponding adversarial examples are more likely to be close in the space of normalized output). By doing this, the other modules (e.g., convolutional or fully connected layer) are easier to learn the native representation of the data (domain-invariant representations).

---

### Official Review · AnonReviewer3 · 2020-10-27
**Good empirical results, some inconsistencies need to be fixed**

**Rating:** 6
**Confidence:** 3

**Review:**

This paper introduces an algorithm for defending against multiple adversarial attacks (L1, L2, L-inf) by learning separate batch-norm statistics for each attack type. At inference time, the batch-normalized outputs corresponding to the different attack types are averaged according to the probability of attack types as given by a separate prediction network (one per batch-norm layer). The algorithm is evaluated on a range of datasets (MNIST, CIFAR-10, Tiny-ImageNet) and is shown to outperform recent competing approaches on a range of adversarial attacks.

Strengths:
- Empirical evaluation covers a range of datasets and attacks, including the most recent strong attacks
- Performance improvement is significant over competing methods

Weaknesses:
- Experiment results seem to be from only a single run
- Proposed method only applies to architectures that use batch-norm, unlike competing methods
- Inconsistencies in the motivation vs empirical analyses, and experimental setups:
  * Plots in Fig 1 to support the multi-domain hypothesis are on VGG16, but this is not the architecture used in experiments (ResNets were used instead)
  * Main experimental results are on ResNets, but ablation studies are on VGG16
  * Ablation study shows that adding GBN to the first layer results in the best performance (Fig 3a) but batch-norm statistics in this layer are almost indistinguishable between the different attack types (Fig 6a,b).

Overall this paper introduces a new approach to defending against multiple adversarial attacks based on batch-norm statistics, with strong improvements over competing methods. Where it falls slightly short is in the inconsistencies mentioned above between motivation, empirical analyses to support the motivation and experimental setups; I will gladly raise my score accordingly if these are fixed.

Other questions and comments:

- What do the batch-norm stats look like on the ResNet architectures used? What is the effect of network architecture on the method?
- How many runs were the reported results dervied from?
- What is the performance of the best performing model (e.g. using adversarial training) under an individual attack? The proposed method almost seems like training separate models for each attack type - how does it perform against such a baseline where you have a classifier to pick the relevant model?
- Why is the algorithm not trained using soft-gating, which will better reflect the actual prediction made at inference time?
- It would be useful to show evidence of "domain invariant representations" learnt, for instance, using tSNE plots of the representations.
- Clarify the meaning of "All attacks" in the results tables
- I'm not sure if the theorem provided in the paper is helpful - the loss function -yf(x) is not commonly used.

*** Post response comments ***
I thank the authors for the clarifications and additional data provided in the revision; most of my concerns have been addressed. I have to say it is a bit worrying that the indistinguishable statistics in the first layer also occurs for the WideResNet models, which needs more explanation and exploration since it is a key motivation of the method. Also, the point raised by R1 on the evaluation is somewhat concerning. As such, I think this paper is borderline even though I am raising my rating to 6.

---

> ### Author Response · Authors · 2020-11-16
> **Clarification of inconsistencies and other questions (1/2)**
>
> We thank the reviewer for appreciating our efforts and providing constructive comments in this COVID-19 outbreak! Stay safe and healthy. We will clarify all the inconsistencies and your concerns in a revised submission soon, so we humbly expect you can reconsider the final decision.
>
>
> ***Q1***: Experiment results seem to be from only a single run:
>
> ***A***: For all attacks except FGSM and APGD$_{CE}$  (the default setting in AutoAttack), we made 5 random restarts for each of the results on different datasets. As for models using our GBN, we trained each model 3 times, and the results are similar, showing that training with GBN is stable. We will provide more details in our revision soon.
>
>
> ***Q2***: Proposed method only applies to architectures that use batch-norm, unlike competing methods:
>
> ***A***: Firstly, BN has been widely used as an essential module for different tasks and model architectures to stabilize and accelerate the training of DNNs. Based on the BN module in this paper, our GBN is applicable to a variety of architectures.
> Secondly, this paper indeed does address a general idea to defend multiple adversarial perturbation types: align the distributions among different perturbations by using one normalization module to handle one perturbation type (learn domain-specific features), and then the aligned distribution contribute to the subsequent layers or modules (e.g., convolutional or fully connected layer) for learning domain-invariant representations.  We only provide a simple way to achieve this goal by using the population statistics of BN, and we believe the extra affine parameters of BN/GN (Group Normalization)/LN (Layer Normalization)/IN (Instance Normalization) can also contribute to it. We believe this general idea, firstly proposed by us, can be widely used on various architectures and tasks, to defend multiple adversarial perturbation types. We will further study it in future work.
>
>
> ***Q3***: Inconsistencies in the motivation vs empirical analyses, and experimental setups:
>
> ***(Q3.1-Q3.2)***  ***A***: We use VGG- 16 instead of ResNet for the hypothesis in Fig 1 and the layer-wise ablation study because VGG has a comparatively simple and linear architecture. Thus, we can eliminate other influence factors (e.g., skip connections in ResNet, dense connections in DenseNet, etc.) and better observe the phenomena. For the main experiments, we choose ResNet since it's one of the most commonly used architecture in the adversarial learning literature (Xie et al., 2018; Liao et al., 2018; Pratyush et al., 2020). Also, we're happy to re-organize the main text and add new experiments (VGG-16) in the revision soon.
>
> ***(Q3.3)*** ***A***: Actually, it is not inconsistent or contradictory between Fig 3a and Fig 6 a,b. We may understand it from three following viewpoints: (a) gated sub-network prediction accuracy. We provided the prediction accuracy of the gated sub-network on inputs from different domains in Table 8 in the appendix. According to the experimental results, the gated sub-network achieves the highest prediction accuracy in the first layer, though the running statistics in the first layer are almost indistinguishable. (b)  the convolutional layers in the gated-subnetwork may still be able to distinguish the differences between these features (we provided some discussions in Appendix C). (c) adding GBN to the first layer shows better performance than solely adding GBN to deeper layers might because those shallow layers are more critical to model robustness (Liu et al. 2019b). We will make it clearer in the revision soon.
>
>
> ***Q4***: What do the batch-norm stats look like on the ResNet architectures used? What is the effect of network architecture on the method?:
>
> ***A***: We will provide the BN statistics of ResNet-20 and show the results of GBN using VGG-16 in the revision soon. As for the effect of network architecture on the method, our method can be applied to different models and is stable among different architectures (LeNet, ResNet-20, ResNet-34).
>
>
> ***Q5***: What is the performance of the best performing model (e.g. using adversarial training) under an individual attack? The proposed method almost seems like training separate models for each attack type - how does it perform against such a baseline where you have a classifier to pick the relevant model?:
>
> ***A***: We will compare with models training solely on one specified perturbation type ($\ell_1$, $\ell_2$, and $\ell_{\infty}$, respectively) in our revision soon.

---

> > ### Author Response · Authors · 2020-11-16
> > **Clarification of inconsistencies and other questions (2/2)**
> >
> > ***Q6***: Why is the algorithm not trained using soft-gating, which will better reflect the actual prediction made at inference time?:
> >
> > ***A***: If we provide soft-gating during training, we will update each BN branch statistics using the mixture of data from different domains. Thus, our implementation is more similar to Mode Normalization (MN) in this way which will fail our motivation. However, MN performs far worse than our GBN in Table 1 a,b,c by large margins.
> > Our key thinking is to align the distributions among different perturbations by using one normalization module to handle one perturbation type (learn domain-specific features), and then the aligned distribution contributes to the subsequent layers or modules (e.g., convolutional or fully connected layer) for learning domain-invariant representations. Thus, we need to capture domain-specific features by forcing data from different domains to go through corresponding BN with hard label supervision during training. We also provided the results of using different prediction approaches for the gated sub-network during inference in Appendix E.5 (hard or soft), and the performance is similar.
> > We will add more discussions and explanations in the revision soon.
> >
> > ***Q7***: It would be useful to show evidence of "domain invariant representations" learnt, for instance, using tSNE plots of the representations:
> >
> > ***A***: Thanks, we will try your suggestions and add related discussions in the revision soon.
> >
> > ***Q8***: Clarify the meaning of "All attacks" in the results tables:
> >
> > ***A***: It represents the worst-case performance computed over all attacks for each example ((Pratyush et al., 2020). Specifically, given an image, if we generate an adversarial attack that leads to wrong predictions using any attacks, we mark it as a failure. Thanks, we will make it clearer in our revision soon.
> >
> > ***Q9***: I'm not sure if the theorem provided in the paper is helpful - the loss function -yf(x) is not commonly used:
> >
> > ***A***: Thanks. Our results here can be easily extended to the case that the loss function can be expressed as $l(f(x), y)=h(-f(x)y)$ where $h$ is a non-decreasing function. This is a broad family of losses including the hinge loss and logistic loss. We're also happy to follow your suggestions and cut the space of the theorem in the main text in the revision soon.

---

### Author Response · Authors · 2020-11-23
**Summary of changes**

We are grateful for the reviewers' constructive feedback! We have revised our submission, and we summarize the major changes below:

(1) In Section 3.1, we cut down the space of the theorem.

(2) In Section 4.1, we briefly describe the attack algorithms of each perturbation type we used, corresponding to the details discussed in Appendix D.1. We add the experiments on defense methods TRADES and PGD adversarial training on single perturbation types ($P_1$, $P_2$, and $P_{\infty}$). We add the results using models VGG-16 and WideResNet-28-10.

(3) In Section 4.2, we explain the definition of 'all attacks accuracy', and present the number of runs for each experiment; we add a new paragraph to discuss the different variants of white-box attacks (fool the GBN layer and generate adversarial examples by manually selecting BN branches); we provide more analysis about the novelty of our GBN, and add more comparison between GBN and MN in the paragraph 'Comparison with other normalization methods'; we also provide the results of additional defenses (TRADES, $P_1$,$P_2$, and $P_{\infty}$) and show the Standard Deviation of 'all attacks' in Table 1.

(4) In Section 4.3, we provide the predictions of the gated sub-network; we cite the paper Asano et al., 2020.

(5) We add the experimental results on CIFAR-10 using VGG-16 and WideResNet-28-10 in Appendix E.3 (Table 7, 8). We show that our GBN still outperforms other methods and can be applied to different model architectures.

(6) We add the experimental results on MNIST and CIFAR-10 of the additional defenses (TRADES, $P_1$, $P_2$, and $P_{\infty}$) in Appendix E.3 (Table 4, 5, 7, 8). Again, we show that our GBN outperforms these additional defense methods.


(7) We add PGD-k attacks with different iteration steps (i.e., k=50, 100, 200, 1000) on CIFAR-10 in Appendix E.4 (Table10); we also update the results of PGD-1000-$\ell_{\infty}$ in Table 5, 7, and 8. The results show that our GBN still outperforms other baselines.

(8) We add the experimental results of adaptive attacks by fooling the GBN layers in Appendix E.8 (Table 15 and 16). The results demonstrate that our GBN remains at the same level of robustness against these attacks which indicates our GBN does not rely on obfuscated gradients.

(9) We add the experimental results of feature visualization before and after GBN blocks using t-SNE in Appendix E.9 (Figure 6 and 7). The results demonstrate the motivation of our GBN to align feature to domain-invariant representations, i.e., one clean example and its corresponding adversarial examples are more likely to be close in the space of normalized output.

(10) We add the running statistics of WideResNet-28-10 with the multiple BN branch structure in Appendix E.10 (Figure 9). The results confirm that different perturbation types induce different normalization statistics.

(11) We carefully correct our reference issues in the revised version.

---

### Decision · Program_Chairs · 2021-01-07
**Final Decision**

**Decision:**

Reject

**Comment:**

This paper first examines a multi-domain separation phenomenon, where different types of adversarial noise lead to different running statistics, and then introduces Gated Batch Normalization (GBN), a building block for deep neural networks that improves robustness against multiple perturbation types. GBN consists of a gated subnetwork and a multi-branch BN layer, and each BN branch handles a single type of perturbation. Results were reported on MNIST, CIFAR-10, and Tiny-ImageNet. Though the idea and methodology are valid and of interest, some concerns regarding the experimental section remain after rebuttal discussions.